# Multicolor fluorescence fluctuation spectroscopy in living cells via spectral detection

**Valentin Dunsing\*, Annett Petrich, Salvatore Chiantia\***

Universität Potsdam, Institute of Biochemistry and Biology, Potsdam, Germany

**Abstract** Signaling pathways in biological systems rely on specific interactions between multiple biomolecules. Fluorescence fluctuation spectroscopy provides a powerful toolbox to quantify such interactions directly in living cells. Cross-correlation analysis of spectrally separated fluctuations provides information about intermolecular interactions but is usually limited to two fluorophore species. Here, we present scanning fluorescence spectral correlation spectroscopy (SFSCS), a versatile approach that can be implemented on commercial confocal microscopes, allowing the investigation of interactions between multiple protein species at the plasma membrane. We demonstrate that SFSCS enables cross-talk-free cross-correlation, diffusion, and oligomerization analysis of up to four protein species labeled with strongly overlapping fluorophores. As an example, we investigate the interactions of influenza A virus (IAV) matrix protein 2 with two cellular host factors simultaneously. We furthermore apply raster spectral image correlation spectroscopy for the simultaneous analysis of up to four species and determine the stoichiometry of ternary IAV polymerase complexes in the cell nucleus.

**\*For correspondence:**
valentin.dunsing@gmx.de (VD);
chiantia@uni-potsdam.de (SC)

**Competing interest:** The authors declare that no competing interests exist.

## Introduction

Living cells rely on transport and interaction of biomolecules to perform their diverse functions. To investigate the underlying molecular processes in the native cellular environment, minimally invasive techniques are needed. Fluorescence fluctuation spectroscopy (FFS) approaches provide a powerful toolbox that fulfills this aim (*Jameson et al., 2009*; *Weidemann et al., 2014*; *Petazzi et al., 2020*). FFS takes advantage of inherent molecular dynamics present in biological systems, for example, diffusion, to obtain molecular parameters from fluctuations of the signal emitted by an ensemble of fluorescent molecules. More in detail, the temporal evolution of such fluctuations allows the quantification of intracellular dynamics. In addition, concentration and oligomerization state of molecular complexes can be determined by analyzing the magnitude of fluctuations. Finally, hetero-interactions of different molecular species can be detected by cross-correlation analysis of fluctuations emitted by spectrally separated fluorophores (*Schwille et al., 1997*). Over the last two decades, several experimental FFS schemes such as raster image (cross-) correlation spectroscopy (RI(C)CS) (*Digman et al., 2005*; *Digman et al., 2009b*), (cross-correlation) Number&Brightness analysis (*Digman et al., 2008*; *Digman et al., 2009a*), and imaging FCS (*Krieger et al., 2015*) have been developed, extending the concept of traditional single-point fluorescence (cross-) correlation spectroscopy (F(C)CS) (*Magde et al., 1972*). A further interesting example of FFS analysis relevant in the field of cell biology is represented by scanning F(C)CS (SF(C)CS). Using a scan path perpendicular to the plasma membrane (PM), this technique provides enhanced stability and the ability to probe slow membrane dynamics (*Ries and Schwille, 2006*), protein interactions (*Ries et al., 2009b*; *Dunsing et al., 2017*), and oligomerization (*Dunsing et al., 2018*) at the PM of cells.

FFS studies are conventionally limited to the analysis of two spectrally distinguished species due to (i) broad emission spectra of fluorophores with consequent cross-talk artifacts and (ii) limited overlap

of detection/excitation geometries for labels with large spectral separation. Generally, only a few fluorescence-based methods are available to detect ternary or higher order interactions of proteins (*Galperin et al., 2004*; *Sun et al., 2010*; *Hur et al., 2016*). First in vitro approaches to perform FCS on more than two species exploited quantum dots (*Burkhardt et al., 2005*) or fluorescent dyes with different Stokes shifts excited with a single laser line in one- (*Hwang et al., 2006*) or two-photon excitation (*Heinze et al., 2004*; *Ridgeway et al., 2012a*), coupled with detection on two or more single photon counting detectors. Following an alternative conceptual approach, it was shown in vitro that two spectrally strongly overlapping fluorophore species can be discriminated in FCS by applying statistical filtering of detected photons based on spectrally resolved (fluorescence spectral correlation spectroscopy [FSCS]; *Benda et al., 2014*) or fluorescence lifetime (fluorescence lifetime correlation spectroscopy [FLCS]; *Böhmer et al., 2002*; *Kapusta et al., 2007*; *Ghosh et al., 2018*) detection. Such framework allows the minimization of cross-talk artifacts in FCCS measurements performed in living cells (*Padilla-Parra et al., 2011*). Recently, three-species implementations of RICCS and FCCS were successfully demonstrated for the first time in living cells. Schrimpf et al. presented raster spectral image correlation spectroscopy (RSICS), a powerful combination of RICS with spectral detection and statistical filtering based on the emission spectra of mEGFP, mVenus, and mCherry fluorophores (*Schrimpf et al., 2018*). Stefl et al. developed single-color fluorescence lifetime cross-correlation spectroscopy (sc-FLCCS), taking advantage of several GFP variants characterized by short or long fluorescence lifetimes (*Štefl et al., 2020*). Using this elegant approach, three-species FCCS measurements could be performed in yeast cells, with just two excitation lines.

Here, we explore the full potential of FSCS and RSICS. In particular, we present scanning fluorescence spectral correlation spectroscopy (SFSCS), combining SFCS and FSCS. We show that SFSCS enables cross-talk-free SFCCS measurements of two protein species at the PM of living cells tagged with strongly overlapping fluorophores in the green or red regions of the visible spectrum, excited with a single excitation line. This approach results in correct estimates of protein diffusion dynamics, oligomerization, and interactions between both species. Further, we extend our approach to the analysis of three or four interacting partners: by performing cross-correlation measurements on different fluorescent protein (FP) hetero-oligomers, we demonstrate that up to four FP species can be simultaneously analyzed. We then apply this scheme to simultaneously investigate the interaction of influenza A virus (IAV) matrix protein 2 (M2) with two cellular host factors, the tetraspanin CD9 and the autophagosome protein LC3, co-expressed in the same cell. Finally, we extend RSICS for the detection of four molecular species and quantify, for the first time directly in living cells, the complete stoichiometry of ternary IAV polymerase complexes assembling in the nucleus, using three-species fluorescence correlation and brightness analysis.

## Results
### Cross-talk-free SFSCS analysis of membrane-associated proteins using FPs with strongly overlapping emission spectra and a single excitation wavelength

To test the suitability of SFSCS to quantify interactions between membrane proteins tagged with strongly spectrally overlapping fluorophores, we investigated HEK 293T cells co-expressing myristoylated and palmitoylated mEGFP (mp-mEGFP) and mp-mEYFP. These monomeric FPs are anchored independently to the inner leaflet of the PM and their emission maxima are only ca. 20 nm apart (*Figure 1—figure supplement 1*). The signal originating from the two fluorophores was decomposed using spectral filters (*Figure 1—figure supplement 2A*) based on the emission spectra detected on cells expressing mp-mEGFP and mp-mEYFP separately (*Figure 1—figure supplement 1*). We then calculated autocorrelation functions (ACFs) and the cross-correlation function (CCF) for signal fluctuations assigned to each fluorophore species. Representative CFs for a typical measurement are shown in *Figure 1A*, indicating absence of interactions and negligible cross-talk between the two FPs. In contrast, we observed substantial CCFs when analyzing measurements on cells expressing mp-mEYFP-mEGFP heterodimers (*Figure 1—figure supplement 3A*). Overall, we obtained a relative cross-correlation (rel.cc.) of 0.72 ± 0.12 (mean ± SD, n = 22 cells) in the latter sample compared to a vanishing rel.cc. of 0.02 ± 0.04 (mean ± SD, n = 34 cells) in the negative control (*Figure 1B*). Comparison of two types of linker peptides (short flexible or long rigid) between mEGFP and mEYFP

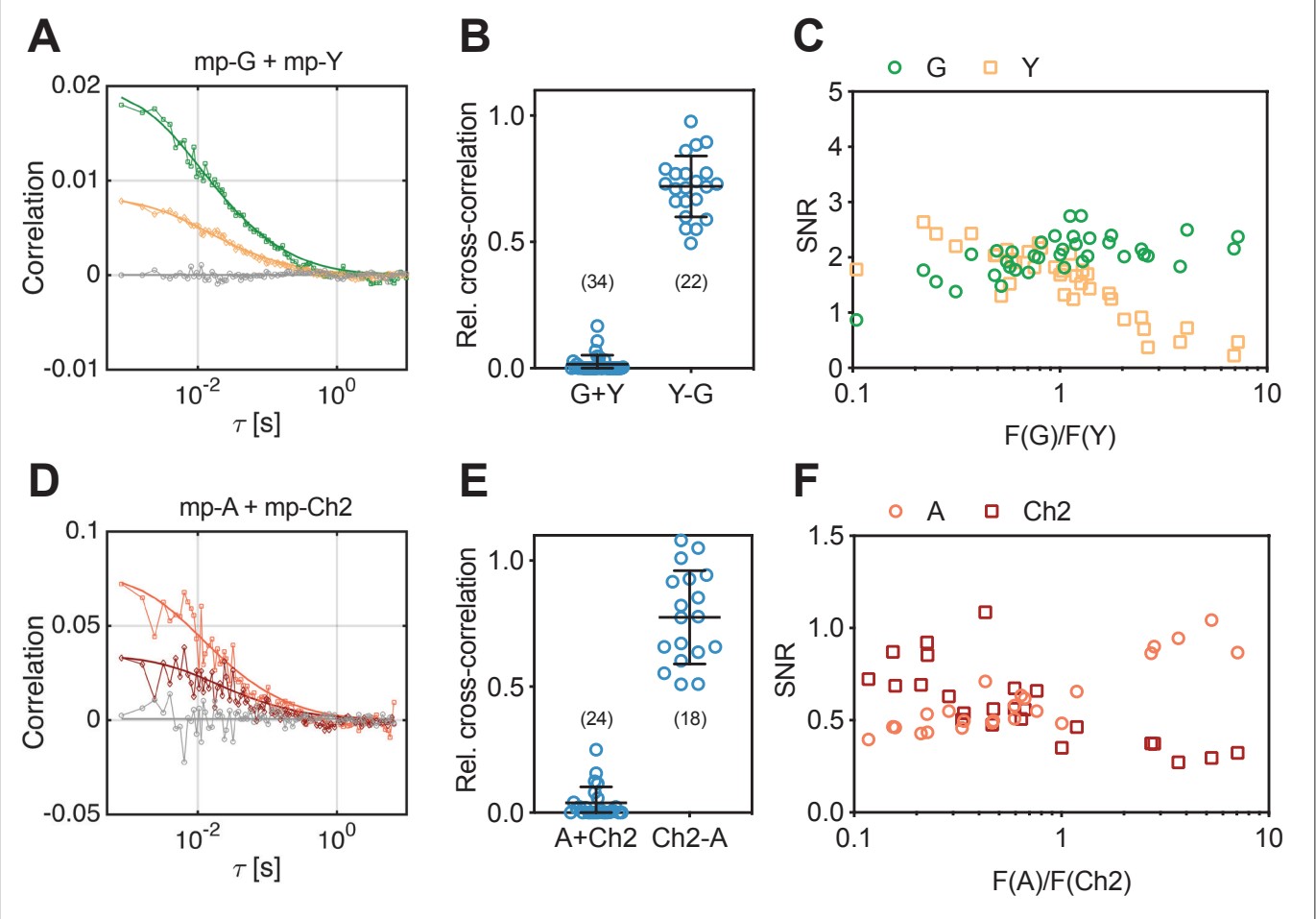

**Figure 1.** Cross-correlation and signal-to-noise ratio (SNR) analysis for two-species scanning fluorescence spectral correlation spectroscopy (SFSCS) measurements at the plasma membrane (PM) of HEK 293T cells, performed with fluorescent proteins (FPs) showing strongly overlapping emission spectra. (**A**) Representative correlation functions (CFs) (green: autocorrelation function [ACF] for mEGFP ['G']; yellow: ACF for mEYFP ['Y'] gray: cross-correlation function [CCF] calculated for both fluorophore species) obtained from SFSCS measurements on the PM of HEK 293T cells co-expressing mp-mEGFP and mp-mEYFP. Solid thick lines show fits of a two-dimensional diffusion model to the CFs. (**B**) Relative cross-correlation values obtained from SFSCS measurements described in (**A**) ('G + Y') or on HEK 293T cells expressing mp-mEYFP-mEGFP heterodimers ('Y-G'). (**C**) SNR of ACFs for mEGFP (green) and mEYFP (yellow), obtained from SFSCS measurements described in (**A**), plotted as a function of the average ratio of detected mEGFP and mEYFP fluorescence. (**D**) Representative CFs (light red: ACF for mApple ['A']; dark red: ACF for mCherry2 ['Ch2']; gray: CCF calculated for both fluorophores) obtained from SFSCS measurements on the PM of HEK 293T cells co-expressing mp-mApple and mp-mCherry2. Solid thick lines show fits of a two-dimensional diffusion model to the CFs. (**E**) Relative cross-correlation values obtained from SFSCS measurements described in (**D**) ('A + Ch2') or on HEK 293T cells expressing mp-mCherry2-mApple heterodimers ('Ch2-A'). (**F**) SNR of ACFs for mApple (light red) and mCherry2 (dark red), obtained from SFSCS measurements described in (**D**), plotted as a function of the average ratio of detected mApple and mCherry2 fluorescence. Data are pooled from three (**B**) or two (**E**) independent experiments each. The number of cells measured is given in parentheses. Error bars represent mean ± SD.

The online version of this article includes the following source data and figure supplement(s) for figure 1:

**Source data 1.** Relative cross-correlation and signal-to-noise ratios for two-species scanning fluorescence correlation spectroscopy measurements.

**Figure supplement 1.** Fluorescent protein (FP) emission spectra.

**Figure supplement 2.** Spectral filters for two-species scanning fluorescence spectral correlation spectroscopy (SFSCS).

**Figure supplement 3.** Scanning fluorescence spectral correlation spectroscopy (SFSCS) on fluorescent protein (FP) heterodimers.

showed that the linker length slightly affected rel.cc. values obtained for heterodimers (*Figure 1—figure supplement 3C*). FPs linked by a short peptide displayed lower rel.cc., probably due to fluorescence resonance energy transfer (FRET), as previously reported (*Foo et al., 2012*). Therefore, unless otherwise noted, similar long rigid linkers were inserted in all constructs used in this study that contain multiple FPs (see *Supplementary file 1a*).

Overlapping fluorescence emission from different species detected in the same channels provides unwanted background signal and thus reduces the signal-to-noise ratio (SNR) of the CFs (*Schrimpf et al., 2018*). To assess to which extent the SNR depends on the relative concentration of mEGFP and mEYFP fluorophores, we compared it between measurements on cells with different relative expression levels of the two membrane constructs (*Figure 1C*). While the SNR of mEGFP ACFs was only moderately affected by the presence of mEYFP signal (i.e., SNR ranging from ca. 2.5 to 1.0, with 90% to 10% of the signal originating from mEGFP), the ACFs measured for mEYFP showed strong noise when mEGFP was present in much higher amount (i.e., SNR ranging from 2.5 to 0.2, with 90% to 10% of the signal originating from mEYFP).

Next, we tested whether the same approach can be used for FPs with overlapping emission in the red region of the visible spectrum, which generally suffer from reduced SNR in FFS applications (*Dunsing et al., 2018*; *Foust et al., 2019*). Therefore, we performed SFSCS measurements on HEK 293T cells co-expressing mp-mCherry2 and mp-mApple. Also, the emission spectra of these FPs are shifted by less than 20 nm (*Figure 1—figure supplement 1*, spectral filters are shown in *Figure 1—figure supplement 2B*). Correlation analysis resulted generally in noisier CFs (*Figure 1D*) compared to mEGFP and mEYFP. Nevertheless, a consistently negligible rel.cc. of 0.04 ± 0.06 (mean ± SD, n = 24 cells) was observed. In contrast, a high rel.cc. of 0.78 ± 0.19 (mean ± SD, n = 18 cells) was obtained on cells expressing mp-mCherry2-mApple heterodimers (*Figure 1E*, *Figure 1—figure supplement 3B*). SNR analysis confirmed lower SNRs of the CFs obtained for red FPs (*Figure 1F*) compared to mEGFP and mEYFP, with values for mApple depending more weakly on the relative fluorescence signal than mCherry2 (i.e., ca. twofold change for mApple vs. ca. fourfold change for mCherry2, when the relative abundance changed from 90% to 10%).

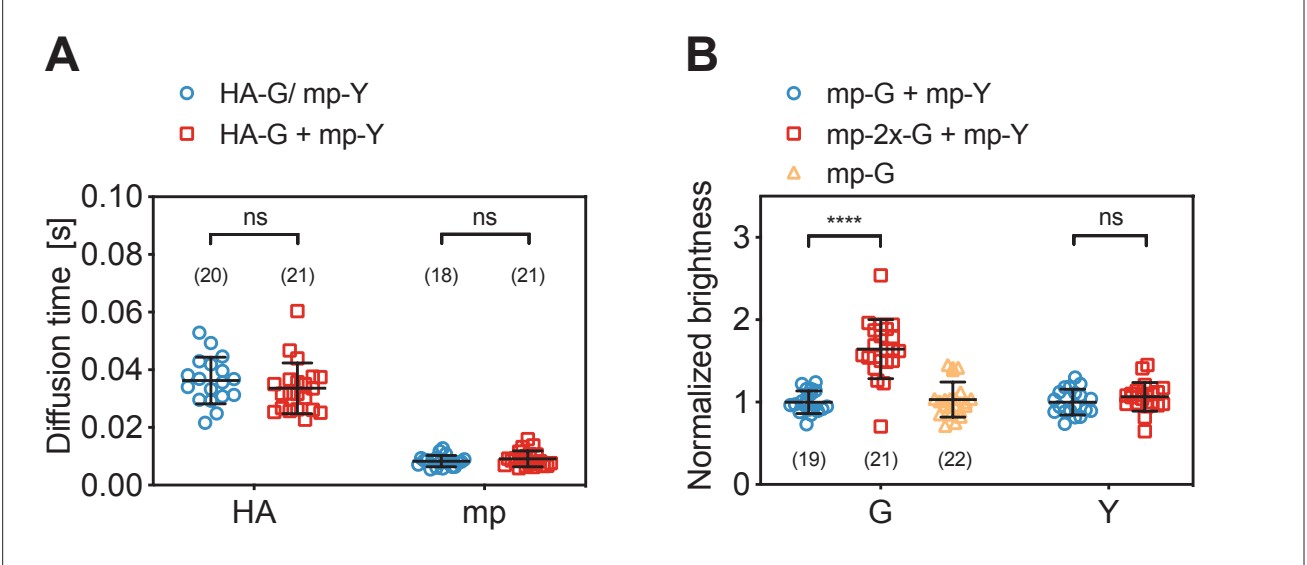

**Figure 2.** Diffusion and molecular brightness analysis for two-species scanning fluorescence spectral correlation spectroscopy (SFSCS) measurements at the plasma membrane (PM) of HEK 293T cells. (**A**) Diffusion times obtained from SFSCS measurements on HEK 293T cells expressing either influenza A virus (IAV) HA-mEGFP or mp-mEYFP separately (blue), or co-expressing both fusion proteins (red). (**B**) Normalized molecular brightness values obtained from SFSCS measurements on HEK 293T cells co-expressing mp-mEGFP and mp-mEYFP (blue), mp-2x-mEGFP and mp-mEYFP (red), or expressing mp-mEGFP alone (yellow). Normalized brightness values were calculated by dividing molecular brightness values detected in each SFSCS measurement by the average brightness obtained for mEGFP and mEYFP in cells co-expressing mp-mEGFP and mp-mEYFP. Data are pooled from two independent experiments for each sample. The number of cells measured is given in parentheses. Error bars represent mean ± SD. Statistical significance was determined using Welch's corrected two-tailed Student's *t*-test (****p<0.0001, ns: not significant).

The online version of this article includes the following source data and figure supplement(s) for figure 2:

**Source data 1.** Diffusion times and normalized molecular brightness values for two-species scanning fluorescence correlation spectroscopy measurements.

**Figure supplement 1.** Relative cross-correlation obtained from two-species scanning fluorescence spectral correlation spectroscopy (SFSCS) measurements.

We furthermore verified that SFSCS analysis results in correct estimates of protein diffusion dynamics. To this aim, we co-expressed mEGFP-tagged IAV hemagglutinin spike transmembrane protein (HA-mEGFP) and mp-mEYFP. We then compared the diffusion times measured by SFSCS to the values obtained on cells expressing each of the two constructs separately (*Figure 2A*). For HA-mEGFP, an average diffusion time of 34 ± 9 ms (mean ± SD, n = 21 cells) was determined in cells expressing both proteins. This value was comparable to that measured for HA-mEGFP expressed separately (36 ± 8 ms, mean ± SD, n = 18 cells). For mp-mEYFP, diffusion times of 9 ± 3 ms and 8 ± 2 ms were measured in samples expressing both proteins or just mp-mEYFP, respectively. In addition to diffusion analysis, we also analyzed the cross-correlation of HA-mEGFP and mp-mEYFP signal for two-species measurements, resulting in negligible rel.cc. values (*Figure 2—figure supplement 1*). Hence, SFSCS yielded correct estimates of diffusion dynamics and allowed to distinguish faster and slower diffusing protein species tagged with spectrally strongly overlapping FPs.

Finally, we evaluated the capability of SFSCS to precisely determine the molecular brightness as a measure of protein oligomerization. We compared the molecular brightness values for mEGFP and mEYFP in samples co-expressing monomeric FP constructs mp-mEGFP and mp-mEYFP with the values obtained for cells co-expressing mp-2x-mEGFP homodimers and mp-mEYFP (*Figure 2B*). From SFSCS analysis of measurements in the latter sample, we obtained a normalized molecular brightness of 1.64 ± 0.36 (mean ± SD, n = 21 cells) for mp-2x-mEGFP, relative to the brightness determined in the monomer sample (n = 19 cells). This value is in agreement with our previous quantification of the relative brightness of mEGFP homodimers, corresponding to a fluorescence probability ($p_f$) of ca. 60–75% for mEGFP (*Dunsing et al., 2018*). The $p_f$ is an empirical, FP-specific parameter that was previously characterized for multiple FPs (*Dunsing et al., 2018*). It quantifies the fraction of non-fluorescent FPs due to photophysical processes, such as transitions to long-lived dark states, or slow FP maturation and needs to be taken into account to correctly determine the oligomerization state of FP tagged protein complexes. As a reference for the absolute brightness, we also determined the relative molecular brightness of mEGFP in cells expressing mp-mEGFP alone, yielding a value of 1.03 ± 0.21 (mean ± SD, n = 22 cells). Additionally, the brightness values determined for mEYFP in both two-species samples were similar, with a relative ratio of 1.07 ± 0.18, as expected. This confirms that reliable brightness values were obtained and that dimeric and monomeric species can be correctly identified.

In summary, these results demonstrate that SFSCS analysis of fluorescence fluctuations successfully separates the contributions of FPs exhibiting strongly overlapping emission spectra, yielding correct quantitative estimates of protein oligomerization and diffusion dynamics.

## Simultaneous cross-correlation and brightness analysis for three spectrally overlapping FPs at the PM

In the previous section, we showed that SFSCS enables cross-talk-free cross-correlation analysis of two fluorescent species excited with a single laser line, even in the case of strongly overlapping emission spectra. To explore the full potential of SFSCS, we extended the approach to systems containing three spectrally overlapping fluorophores. We excited mEGFP, mEYFP, and mCherry2 with 488 nm and 561 nm lines simultaneously and detected their fluorescence in 23 spectral bins in the range of 491–695 nm. We measured individual emission spectra (*Figure 1—figure supplement 1*) for single-species samples to calculate three-species spectral filters (*Figure 3—figure supplement 1*), which we then used to decompose the signal detected in cells expressing multiple FPs into the contribution of each species.

As a first step, we performed three-species SFSCS measurements on HEK 293T cells co-expressing mp-mEYFP with either (i) mp-mEGFP and mp-mCherry2 (mp-G+ mp-Y + mp-Ch2) or (ii) mp-mCherry2-mEGFP heterodimers (mp-Ch2-G + mp-Y ). Additionally, we tested a sample with cells expressing mp-mEYFP-mCherry2-mEGFP heterotrimers (mp-Y-Ch2-G). We then calculated ACFs for all three FP species and CCFs for all fluorophore combinations, respectively. In the first sample (mp-G + mp-Y + mp-Ch2), in which all three FPs are anchored independently to the PM, we obtained CCFs fluctuating around zero for all fluorophore combinations, as expected (*Figure 3A*). In the second sample (mp-Ch2-G + mp-Y ), a substantial cross-correlation was detected between mEGFP and mCherry2, whereas the other two combinations resulted in CCFs fluctuating around zero (*Figure 3B*). In the heterotrimer sample, CCFs with low level of noise and amplitudes significantly above zero were successfully obtained for all three fluorophore combinations (*Figure 3C*). From the amplitude ratios

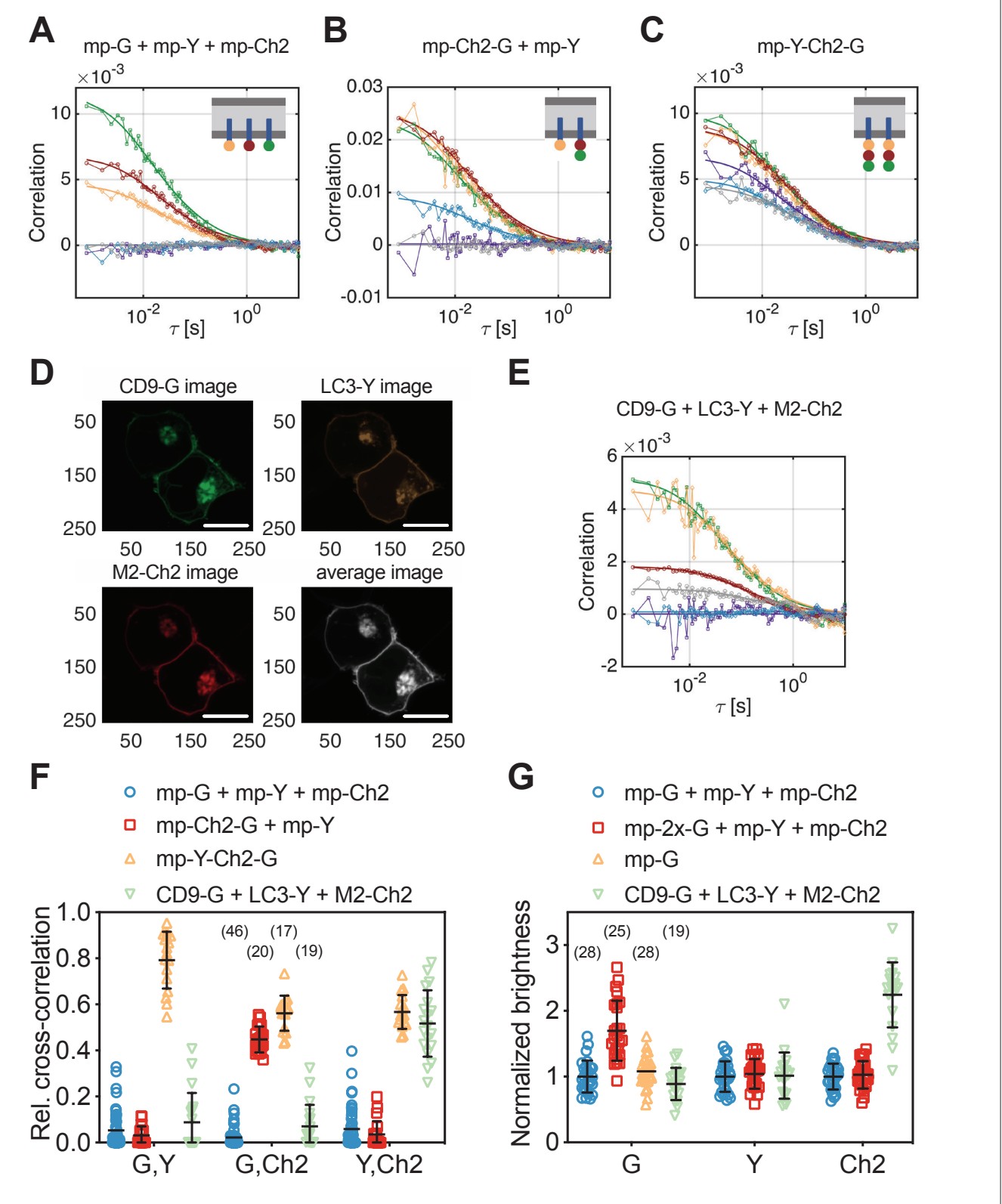

**Figure 3.** Cross-correlation and molecular brightness analysis for three-species scanning fluorescence spectral correlation spectroscopy (SFSCS) measurements on fluorescent protein (FP) hetero-oligomers and influenza A virus (IAV) M2 at the plasma membrane (PM) of HEK 293T cells. (**A–C**) Representative correlation functions (CFs) (green/yellow/red: autocorrelation functions [ACFs] for mEGFP ['G']/mEYFP ['Y']/mCherry2 ['Ch2']; purple/blue/gray: cross-correlation functions [CCFs] calculated for the pairs mEGFP and mEYFP/mEGFP and mCherry2/mEYFP and mCherry2) obtained from

*Figure 3 continued on next page*

*Figure 3 continued*

three-species SFSCS measurements on HEK 293T cells co-expressing mp-mEGFP, mp-mEYFP, and mCherry2 (**A**), mp-mCherry2-mEGFP heterodimers and mp-mEYFP (**B**), or expressing mp-mEYFP-mCherry2-mEGFP heterotrimers (**C**), as illustrated in insets. Solid thick lines show fits of a two-dimensional diffusion model to the CFs. (**D**) Representative fluorescence images of HEK 293T cells co-expressing CD9-mEGFP, LC3-mEYFP, and IAV protein M2-mCh2. Spectral filtering and decomposition were performed to obtain a single image for each species. Scale bars are 5 μm. (**E**) Representative CFs (green/yellow/red: ACFs for mEGFP/mEYFP/mCherry2; purple/blue/gray: CCFs calculated for the pairs mEGFP and mEYFP/mEGFP and mCherry2/ mEYFP and mCherry2) obtained from three-species SFSCS measurements on HEK 293T cells co-expressing CD9-mEGFP, LC3-mEYFP, and M2-mCh2. Solid thick lines show fits of a two-dimensional diffusion model to the CFs. (**F**) Relative cross-correlation values obtained from three-species SFSCS measurements described in (**A–C**) and (**E**). (**G**) Normalized molecular brightness values obtained from three-species SFSCS measurements on HEK 293T cells co-expressing mp-mEGFP, mp-mEYFP, and mp-mCherry2 (blue), mp-2x-mEGFP, mp-mEYFP, and mp-mCherry2 (red), CD9-mEGFP, LC3-mEYFP, and M2-mCh2 (green), or expressing mp-mEGFP alone (yellow). Normalized brightness values were calculated by dividing the molecular brightness values detected in each SFSCS measurement by the average brightness obtained for mEGFP, mEYFP, and mCherry2 in cells co-expressing mp-mEGFP, mp-mEYFP, and mp-mCherry2. Data are pooled from two independent experiments for each sample. The number of cells measured is given in parentheses. Error bars represent mean ± SD.

The online version of this article includes the following source data and figure supplement(s) for figure 3:

**Source data 1.** Relative cross-correlation and normalized molecular brightness values for three-species scanning fluorescence correlation spectroscopy measurements.

**Figure supplement 1.** Spectral filters for three-species scanning fluorescence spectral correlation spectroscopy (SFSCS).

**Figure supplement 2.** Relative cross-correlation (rel.cc.) for three-species scanning fluorescence spectral correlation spectroscopy (SFSCS) analyzed using different fitting algorithms.

**Figure supplement 3.** Noise analysis of three-species scanning fluorescence spectral correlation spectroscopy (SFSCS) measurements.

**Figure supplement 4.** Membrane recruitment of LC3 in M2-expressing cells.

of the ACFs and CCFs, we then calculated rel.cc. values for all measurements (*Figure 3F*). Low rel.cc. values were obtained for all fluorophore combinations that were not expected to show interactions, for example, 0.05 ± 0.08 (mean ± SD, n = 46 cells) between mEGFP and mEYFP signal in the first sample. It is worth noting that these values, albeit consistently negligible, appear to depend on the specific fitting procedure (see *Figure 3—figure supplement 2* and Materials and methods for details). For mEGFP and mCherry2, similar rel.cc. values of 0.45 ± 0.06 (mean ± SD, n = 20 cells) and 0.56 ± 0.08 (mean ± SD, n = 17 cells) were observed in cells expressing mp-mCherry2-mEGFP heterodimers or mp-mEYFP-mCherry2-mEGFP heterotrimers. The minor difference could be attributed, for example, to different linker peptides (i.e., long rigid linker between FPs in heterotrimers and a short flexible linker in heterodimers), increasing the degree of FRET between mEGFP and mCherry2 in heterodimers and reducing the cross-correlation. The heterotrimer sample showed high rel.cc. values also for the other two fluorophore combinations: mEGFP and mEYFP (rel.cc.$_{G,Y}$ = 0.79 ± 0.12) or mCherry2 and mEYFP (rel.cc.$_{Y,Ch2}$ = 0.57 ± 0.07).

In addition to cross-correlation analysis, we performed molecular brightness measurements on samples containing three FP species. In particular, we compared molecular brightness values obtained by SFSCS on HEK 293T cells co-expressing homodimeric mp-2x-mEGFP, mp-mEYFP, and mp-mCherry2 (mp-2x-G + mp-Y + mp-Ch2) to the values measured on cells co-expressing the three monomeric constructs mp-mEGFP, mp-mEYFP, and mp-mCherry2 (mp-G + mp-Y + mp-Ch2). Whereas similar brightness values were obtained for mEYFP and mCherry2 in both samples, for example, relative brightness of 1.04 ± 0.23 for mEYFP and 1.03 ± 0.21 for mCherry2 (mean ± SD, n = 25 cells/n = 28 cells), a higher brightness of 1.70 ± 0.46 was measured for mEGFP in the first sample (*Figure 3G*). This value corresponds to a $p_f$ of ca. 70% for mEGFP, as expected (*Dunsing et al., 2018*). To confirm that absolute brightness values are not influenced by the spectral decomposition, we also determined the brightness of mEGFP in cells expressing mp-mEGFP alone (*Figure 3G*), resulting in values close to 1 (1.08 ± 0.23, mean ± SD, n = 28 cells).

## The IAV protein M2 interacts strongly with LC3 but not with CD9

Having demonstrated the capability of SFSCS to successfully quantify protein interactions and oligomerization, even in the case of three FPs with overlapping emission spectra, we applied this approach in a biologically relevant context. In more detail, we investigated the interaction of IAV channel protein M2 with the cellular host factors CD9 and LC3. CD9 belongs to the family of tetraspanins and is supposedly involved in virus entry and virion assembly (*Florin and Lang, 2018*; *Hantak et al.,*

*2019*; *Dahmane et al., 2019*). The autophagy marker protein LC3 was recently shown to be recruited to the PM in IAV-infected cells (see also *Figure 3—figure supplement 4A,B*), promoting filamentous budding and virion stability, thus indicating a role of LC3 in virus assembly (*Beale et al., 2014*). To detect hetero-interactions between CD9, LC3, and M2, we co-expressed the fluorescent fusion proteins CD9-mEGFP, LC3-mEYFP, and M2-mCherry2 (i.e., M2 carrying an mCherry2 tag at the extracellular terminus) in HEK 293T cells (*Figure 3D*) and performed three-species SFSCS measurements at the PM (*Figure 3E*).

We then calculated rel.cc. values to quantify pair-wise interactions of the three proteins (*Figure 3F*). The obtained rel.cc. values for CD9-mEGFP with LC3-mEYFP or M2-mCherry2 (rel.cc.$_{CD9-G,LC3-Y}$ = 0.09 ± 0.13, rel.cc.$_{CD9-G,M2-Ch2}$ = 0.07 ± 0.09, mean ± SD, n = 19 cells) were similar to those of the negative cross-correlation control (i.e., cells co-expressing mp-mEGFP, mp-mEYFP, and mp-mCherry2, see previous paragraph). In contrast, we detected a substantial rel.cc. of 0.52 ± 0.14 for LC3-mEYFP and M2-mCherry2. This value was close (ca. 90% on average) to that obtained for this fluorophore combination in measurements on FP heterotrimers, suggesting very strong association of LC3-mEYFP with M2-mCherry2. We furthermore analyzed the molecular brightness for each species, normalized to the monomeric references (*Figure 3G*). While CD9-mEGFP and LC3-mEYFP showed normalized brightness values close to 1 ($B_{CD9-G}$ = 0.89 ± 0.25, $B_{LC3-Y}$ = 1.02 ± 0.35), suggesting that both proteins are monomers, we observed significantly higher relative brightness values for M2-mCherry2 ($B_{M2-Ch2}$ = 2.24 ± 0.49). Assuming a $p_f$ of ca. 60% for mCherry2 (*Dunsing et al., 2018*), the determined relative brightness corresponds to an oligomerization state of $\varepsilon_{M2-Ch2}$ = 3.1 ± 0.8, that is, formation of M2 dimers to tetramers at the PM.

## SFSCS allows simultaneous analysis of protein-protein interactions for four spectrally overlapping FP species

Having demonstrated robust three-species cross-correlation analysis, we aimed to further explore the limits of SFSCS. We investigated therefore whether SFSCS can discriminate differential interactions between four species using the spectral emission patterns of mEGFP, mEYFP, mApple, and mCherry2 for spectral decomposition (*Figure 1—figure supplement 1*, *Figure 4—figure supplement 1*). As a proof of concept, we performed four-species measurements on three different samples: (i) cells co-expressing all four FPs independently as membrane-anchored proteins (mp-G + mp-Y + mp-A + mp-Ch2), (ii) cells co-expressing mp-mCherry2-mEGFP heterodimers, mp-mEYFP, and mp-mApple (mp-Ch2-G + mp-Y + mp-A ), and (iii) cells expressing mp-mEYFP-mCherry2-mEGFP-mApple heterotetramers (mp-Y-Ch2-G-A). We then calculated four ACFs, six CCFs, and rel.cc. values from the amplitude ratios of the ACFs and CCFs. For all fluorophore species, ACFs with amplitudes significantly above zero were obtained. ACFs calculated for mEGFP and mEYFP were characterized by a higher SNR compared to those for the red FPs mApple and, in particular, mCherry2 (*Figure 4A–C*). Nevertheless, reasonable diffusion time values could be determined for all species, showing the largest variation for mCherry2 (*Figure 4—figure supplement 2*).

Noise levels of the CCFs were moderate (*Figure 4D–F*), yet allowing robust fitting and estimation of cross-correlation amplitudes. Based on the determined rel.cc. values (*Figure 4G*), the different samples could successfully be discriminated. In the first sample (mp-G + mp-Y + mp-A + mp-Ch2), negligible to very low values were obtained, that is, at maximum 0.11 ± 0.11 (mean ± SD, n = 12 cells) for mApple and mCherry2. In the second sample (mp-Ch2-G + mp-Y + mp-A ), similarly low rel.cc. values were obtained for all fluorophore combinations, for example, 0.10 ± 0.10 (mean ± SD, n = 13 cells) for mApple and mCherry2, with the exception of mEGFP and mCherry2, showing an average value of 0.55 ± 0.13. For the hetero-tetramer sample, high rel.cc. values were measured for all fluorophore combinations, ranging from 0.42 ± 0.07 (mean ± SD, n = 15 cells) for mEGFP and mApple to 0.78 ± 0.08 for mEGFP and mEYFP. Notably, a significant rel.cc. of 0.53 ± 0.10 was also determined for mApple and mCherry2 signals, that is, from the CCFs exhibiting the lowest SNR.

## RSICS can be extended to simultaneous detection of four fluorophore species

Having identified a set of FPs that is compatible with four-species SFSCS, we aimed to extend the recently presented RSICS method (*Schrimpf et al., 2018*) to applications with four fluorophore species being detected simultaneously. To test the effectiveness of this approach, we carried out

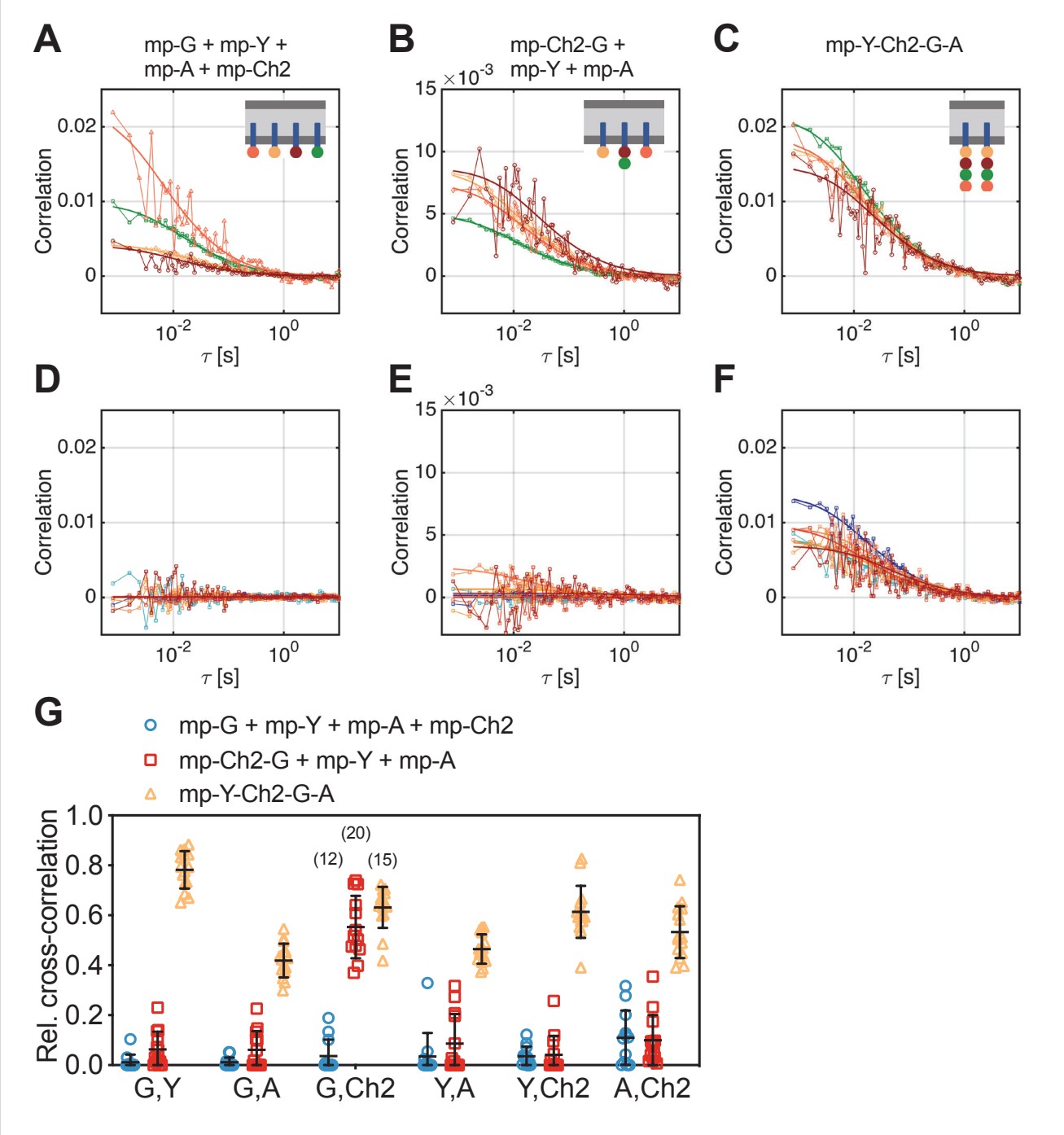

**Figure 4.** Cross-correlation analysis for four-species scanning fluorescence spectral correlation spectroscopy (SFSCS) measurements on fluorescent protein (FP) hetero-oligomers in HEK 293T cells. (**A–C**) Representative autocorrelation functions (ACFs) (green/yellow/orange/red for mEGFP ['G']/mEYFP ['Y']/mApple ['A']/mCherry2 ['Ch2']) obtained from four-species SFSCS measurements on HEK 293T cells co-expressing mp-mEGFP, mp-mEYFP, mp-mApple, and mp-mCherry2 (**A**), mp-mCherry2-mEGFP heterodimers, mp-mEYFP, and mp-mApple (**B**), or expressing mp-mEYFP-mCherry2-mEGFP-mApple hetero-tetramers (**C**), as illustrated in insets. Solid thick lines show fits of a two-dimensional diffusion model to the correlation functions (CFs). (**D–F**) SFSCS cross-correlation functions (CCFs) (dark blue/ light blue/orange/yellow/red/dark red for CCFs calculated for mEGFP and mEYFP/mEGFP and mApple/mEGFP and mCherry2/mEYFP and mApple/mEYFP and mCherry2/mApple and mCherry2) from measurements described in (**A–C**) (CCFs in (**D**)/(**E**)/(**F**)) corresponding to ACFs shown in (**A**)/(**B**)/(**C**). Solid thick lines show fits of a two-dimensional diffusion model to the CFs. (**G**) Relative cross-correlation values obtained from four-species SFSCS measurements described in (**A–C**). Data are pooled from two independent experiments. The number of cells measured is given in parentheses. Error bars represent mean ± SD.

*Figure 4 continued on next page*

*Figure 4 continued*

The online version of this article includes the following source data and figure supplement(s) for figure 4:

**Source data 1.** Relative cross-correlation values for four-species scanning fluorescence correlation spectroscopy measurements.

**Figure supplement 1.** Spectral filters for four-species scanning fluorescence spectral correlation spectroscopy (SFSCS).

**Figure supplement 2.** Diffusion dynamics of four-species scanning fluorescence spectral correlation spectroscopy (SFSCS) measurements.

measurements in the cytoplasm of living A549 cells co-expressing mEGFP, mEYFP, mApple, and mCherry2 in several configurations, similar to the SFSCS experiments presented in the previous paragraph. In more detail, we performed four-species RSICS measurements on the following three samples: (i) cells co-expressing free mEGFP, mEYFP, mApple, and mCherry2 (1x-G + 1x-Y + 1x-A + 1x-Ch2), (ii) cells co-expressing mCherry2-mEGFP and mEYFP-mApple heterodimers (Ch2-G + Y -A), and (iii) cells expressing mEYFP-mCherry2-mEGFP-mApple hetero-tetramers (Y-Ch2-G-A). Representative CFs obtained following RSICS analysis with arbitrary region selection (*Hendrix et al., 2016*) are shown in *Figure 5*. In all samples, ACFs with amplitudes significantly above zero were obtained, with the highest noise level detected for mCherry2 (*Figure 5A, C and E*). A three-dimensional diffusion model could be successfully fitted to all detected ACFs.

Detected CCFs showed the expected pattern: all six CCFs were indistinguishable from noise for the first sample with four independent FPs (*Figure 5B*), whereas large CCF amplitudes were obtained for the pairs mEGFP and mCherry2, as well as mEYFP and mApple in the second sample (Ch2-G + Y- A) (*Figure 5D*). Also, significantly large amplitudes were observed for all six CCFs for the hetero-tetramer sample, albeit with different levels of noise. For example, the lowest SNR was observed in CCFs for mApple and mCherry2 (*Figure 5F*).

From the amplitude ratios of ACFs and CCFs, we determined rel.cc. values (*Figure 5G*). This analysis resulted in negligible values for the first sample (1x-G + 1x-Y + 1x-A + 1x-Ch2): for example, rel. $cc._{G,Ch2} = 0.03 \pm 0.05$ (mean $\pm$ SD, n = 21 cells). For the second sample (Ch2-G + Y -A), values significantly above zero, that is, $rel.cc._{G,Ch2} = 0.46 \pm 0.09$ (mean $\pm$ SD, n = 23 cells) and $rel.cc._{Y,A} = 0.30 \pm 0.10$, were only observed for two fluorophore pairs. For the third sample, cells expressing mEYFP-mCherry2-mEGFP-mApple hetero-tetramers (Y-Ch2-G-A), rel.cc. values significantly above zero were obtained for all FP pairs, ranging from $rel.cc._{A,Ch2} = 0.31 \pm 0.11$ (mean $\pm$ SD, n = 20 cells) to $rel.cc._{G,Y} = 0.60 \pm 0.05$. Notably, rel.cc. values obtained for the FP species correlating in the second sample (Ch2-G + Y -A) were similar for the third sample (Y-Ch2-G-A): for example, $rel.cc._{G,Ch2} = 0.45 \pm 0.07$ and $rel.cc._{Y,A} = 0.41 \pm 0.06$. The lower rel.cc. value measured for mEYFP and mApple in heterodimers (Ch2-G + Y -A) could be attributed to different linker sequences (long rigid linker in heterodimers vs. mCherry2-mEGFP and three long rigid linkers as spacer in hetero-tetramers [Y-Ch2-G-A]), possibly affecting FRET between neighboring FPs.

Finally, we analyzed the diffusion dynamics of FP fusion proteins as determined from the spatial dependence of the ACFs for the four fluorophore species. Diffusion coefficients (D) obtained for mCherry2 showed the highest variation (*Figure 5H*), reflecting the reduced SNR for this fluorophore. Nevertheless, similar average D values were determined for different fluorophore species coupled as hetero-oligomers, for example, $D_G = 19.4 \pm 3.4\ \mu m^2/s$ and $D_{Ch2} = 20 \pm 11\ \mu m^2/s$ (mean $\pm$ SD, n = 23 cells) for mEGFP-mCherry2 heterodimers, and $D_G = 11.2 \pm 2.5\ \mu m^2/s$, $D_Y = 11.6 \pm 2.6\ \mu m^2/s$, $D_A = 12.8 \pm 3.2\ \mu m^2/s$, $D_{Ch2} = 12.6 \pm 5.0\ \mu m^2/s$ (mean $\pm$ SD, n = 20 cells) for hetero-tetramers.

## Cross-correlation and molecular brightness analysis via three-species RSICS provide stoichiometry of IAV polymerase complex assembly

To test the versatility of three-species RSICS, we quantified intracellular protein interactions and stoichiometries in a biologically relevant context. As an example, we focused on the assembly of the IAV polymerase complex (PC), consisting of the three subunits polymerase acidic protein (PA), polymerase basic protein 1 (PB1), and 2 (PB2). A previous investigation using FCCS suggested an assembly model in which PA and PB1 form heterodimers in the cytoplasm of cells. These are imported into the nucleus and appear to interact with PB2 to form heterotrimeric complexes (*Huet et al., 2010*). Nevertheless, the previous analysis could only be performed between two of the three subunits at the same time. Also, the stoichiometry of the complex was reported only for one of the three subunits, that is, PA protein dimerization. Here, we labeled all three subunits using FP fusion constructs and co-expressed

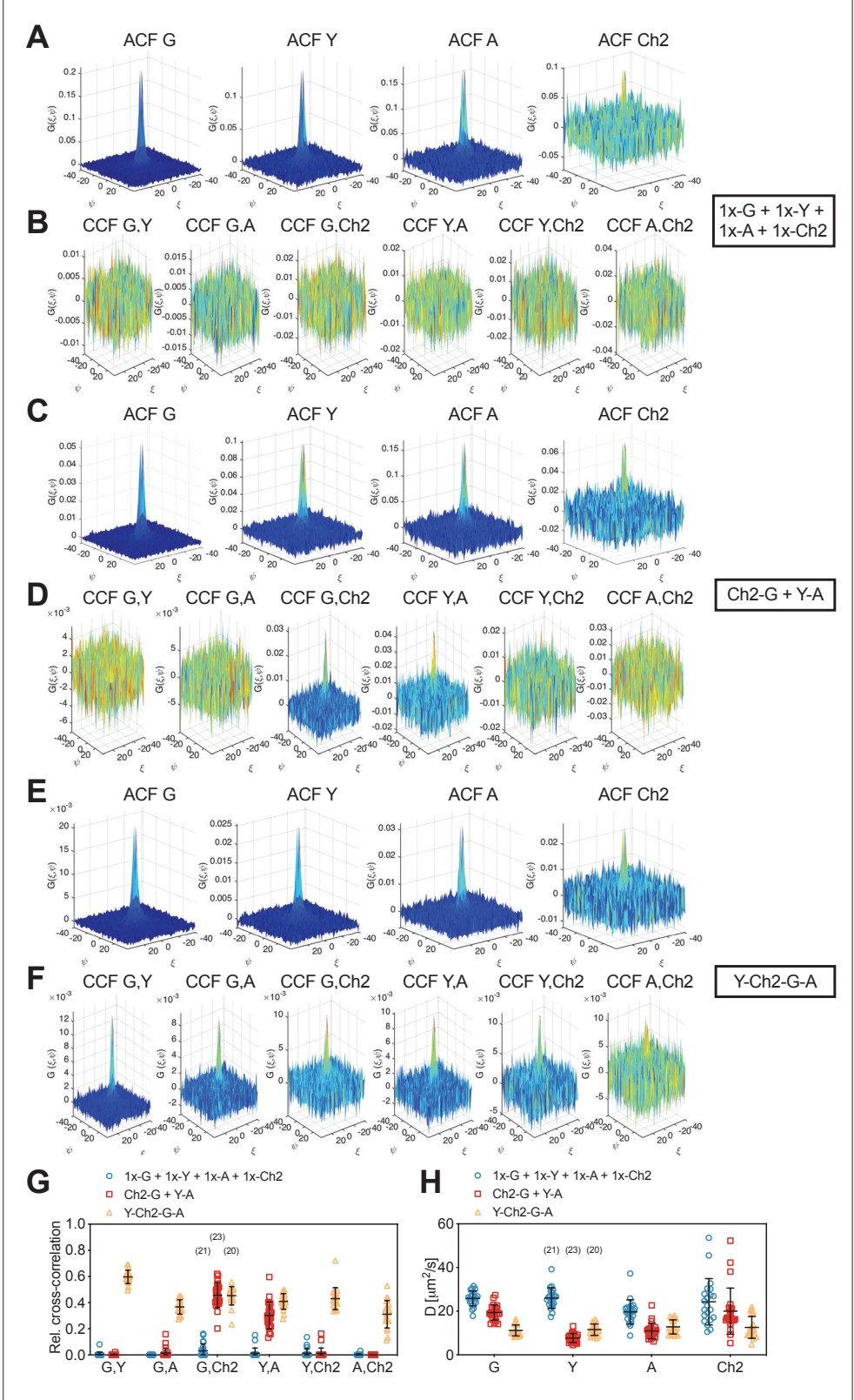

**Figure 5.** Cross-correlation analysis for four-species raster spectral image correlation spectroscopy (RSICS) measurements on fluorescent protein (FP) hetero-oligomers expressed in cytoplasm of A549 cells. (**A–F**) Representative RSICS spatial autocorrelation functions (ACFs) (**A, C, E**) and cross-correlation functions (CCFs) (**B, D, F**) obtained from four-species RSICS measurements on A549 cells. Cells were co-expressing mEGFP ('G'),

*Figure 5 continued on next page*

*Figure 5 continued*

mEYFP ('Y'), mApple ('A'), mCherry2 ('Ch2') (**A, B**), mCherry2-mEGFP and mEYFP-mApple heterodimers (**C, D**), or mEYFP-mCherry2-mEGFP-mApple hetero-tetramers (**E, F**). (**G, H**) Relative cross-correlation values (**G**) and diffusion coefficients (**H**) obtained from four-species RSICS measurements described in (**A–F**). Data are pooled from two independent experiments. The number of cells measured is given in parentheses. Error bars represent mean ± SD.

The online version of this article includes the following source data and figure supplement(s) for figure 5:

**Source data 1.** Relative cross-correlation values and diffusion coefficients for four-species raster spectral image correlation spectroscopy measurements.

**Figure supplement 1.** Fluorescent protein (FP) emission spectra.

**Figure supplement 2.** Fluorescent protein (FP) emission spectra at different pH values.

PA-mEYFP, PB1-mEGFP, and PB2-mCherry2 in A549 cells. We then performed three-species RSICS measurements in the cell nucleus, where all three proteins are enriched (*Figure 6A*). RSICS analysis was performed on an arbitrarily shaped homogeneous region of interest in the nucleus. We then calculated RSICS ACFs (*Figure 6B*), CCFs (*Figure 6C*), and rel.cc. values (*Figure 6D*) for the three fluorophore combinations. The determined rel.cc. values were compared to the values obtained on negative controls (i.e., cells co-expressing free mEGFP, mEYFP, and mCherry) and positive controls (i.e., cells expressing mEYFP-mCherry2-mEGFP heterotrimers) (*Figure 6D*).

For the polymerase sample, high rel.cc. values were observed for all combinations: $\text{rel.cc.}_{\text{PB1-G,PA-Y}}$ = 0.93 ± 0.18 (mean ± SD, n = 53 cells), $\text{rel.cc.}_{\text{PB1-G,PB2-Ch2}}$ = 0.47 ± 0.14, $\text{rel.cc.}_{\text{PA-Y,PB2-Ch2}}$ = 0.39 ± 0.14. For the positive control, similar values were observed for mEGFP and mCherry2, $\text{rel.cc.}_{\text{G,Ch2}}$ = 0.48 ± 0.11 (mean ± SD, n = 46 cells), whereas the values were higher than that measured for PCs for mEYFP and mCherry2, $\text{rel.cc.}_{\text{Y,Ch2}}$ = 0.53 ± 0.11, and lower for mEGFP and mEYFP, $\text{rel.cc.}_{\text{G,Y}}$ = 0.65 ± 0.10. The lower average rel.cc. between PA-mEYFP and PB2-mCherry2 compared to the positive control indicates the presence of a minor fraction of non-interacting PA and PB2. These proteins could be present in the nucleus in unbound form when expressed in higher amount than PB1 since both PA and PB2 localize in the nucleus individually and were previously shown not to interact when both present without PB1 (*Huet et al., 2010*). This explanation is supported by the correlation between rel. $\text{cc.}_{\text{PA-Y,PB2-Ch2}}$ and the relative abundance of PB1-mEGFP (*Figure 6—figure supplement 1A*). Also, the observation that PB1 is only transported to the nucleus in complex with PA is confirmed by the lower concentration of PB1-mEGFP compared to PA-mEYFP in the nuclei of all measured cells (*Figure 6—figure supplement 1A*). Thus, the fraction of PB1-mEGFP bound to PA-mEYFP should be as high as the positive control, for a 1:1 stoichiometry. The observation of higher rel.cc. between mEGFP and mEYFP for the polymerase subunits indicates higher order interactions, that is, higher stoichiometry than 1:1 (*Kaliszewski et al., 2018*).

To quantify the stoichiometry of the PC directly, we analyzed the molecular brightness of RSICS measurements for all three fluorophore species. We normalized the obtained values to the average values determined by RSICS on cells co-expressing monomeric mEGFP, mEYFP, and mCherry2, measured on the same day. To test whether RSICS can be used to obtain reliable brightness/oligomerization values for all fluorophore species, we first performed control experiments on cells co-expressing either (i) 2x-mEGFP homodimers with mEYFP and mCherry monomers (2x-G + 1x-Y + 1x-Ch2) or (ii) the three homodimers 2x-mEGFP, 2x-mEYFP, and 2x-mCherry2 (2x-G + 2x-Y + 2x-Ch2). In the first sample, we observed an increased relative brightness of 1.67 ± 0.38 (mean ± SD, n = 34 cells) for mEGFP, whereas values around 1 were obtained for mEYFP and mCherry2. This confirmed the presence of mEGFP dimers as well as mEYFP and mCherry2 monomers in this control sample, as expected (*Figure 6E*). In the sample containing all three homodimers, increased relative brightness values were observed for all fluorophore species: 1.75 ± 0.37 (mean ± SD, n = 39 cells) for mEGFP, 1.77 ± 0.33 for mEYFP, and 1.61 ± 0.29 for mCherry2 (see *Supplementary file 1b* for data on day-to-day variations). These values indicate successful determination of the dimeric state of all three FP homodimers and are in good agreement with previous brightness measurements on homodimers of mEGFP, mEYFP, and mCherry2, corresponding to $p_f$ values of 60–75% (*Dunsing et al., 2018*). Next, we proceeded with the analysis of PC oligomerization. For each polymerase subunit, relative brightness values close to the values of homodimers were observed. Assuming $p_f$ values of 75, 77, and 61% (as calculated from the determined relative brightness values of homodimers) for mEGFP, mEYFP, and

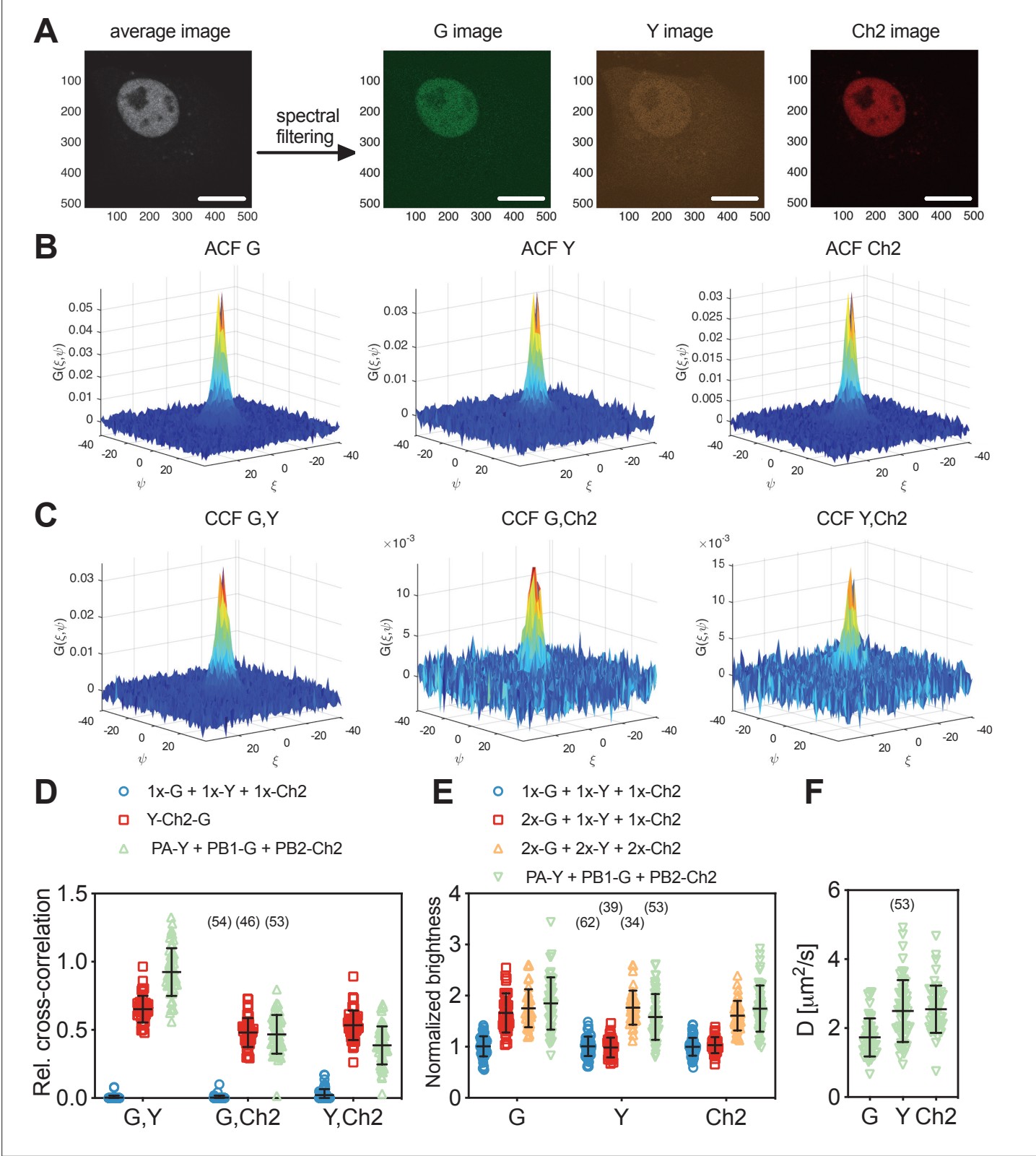

**Figure 6.** Three-species raster spectral image correlation spectroscopy (RSICS) measurements on influenza A virus (IAV) polymerase complex (PC) and fluorescent protein (FP) hetero-oligomers in the nucleus of A549 cells. (**A**) Representative fluorescence image (left) of A549 cells co-expressing FP-tagged IAV PC proteins PA-mEYFP, PB1-mEGFP, and PB2-mCherry2. Spectral filtering and decomposition result in a single image for each species (right), denoted with 'Y,' 'G,' and 'Ch2.' Scale bars are 10 μm. (**B, C**) Representative RSICS spatial autocorrelation functions (ACFs) (**B**) and cross-

*Figure 6 continued on next page*

*Figure 6 continued*

correlation functions (CCFs) (**C**) obtained from three-species RSICS measurements on A549 cells co-expressing PA-mEYFP, PB1-mEGFP, and PB2-mCherry2. (**D**) Relative cross-correlation values obtained from three-species RSICS measurements on A549 cells co-expressing mEGFP, mEYFP, and mCherry2 (blue), PA-mEYFP, PB1-mEGFP, PB2-mCherry2 (green), or expressing mEYFP-mCherry2-mEGFP heterotrimers (red). Data are pooled from four independent experiments. (**E**) Normalized molecular brightness values obtained from three-species RSICS measurements on A549 cells co-expressing mEGFP, mEYFP, and mCherry2 (blue), 2x-mEGFP, mEYFP, and mCherry2 (red), 2x-mEGFP, 2x-mEYFP, 2x-mCherry2 (yellow), or PA-mEYFP, PB1-mEGFP, and PB2-mCherry2 (green). Data are pooled from three (2x-mEGFP + mEYFP + mCherry2, 2x-mEGFP + 2x-mEYFP + 2x-mCherry2), four (PA-mEYFP + PB1-mEGFP + PB2-mCherry2), or five (mEGFP + mEYFP + mCherry2) independent experiments. (**F**) Diffusion coefficients obtained from three-species RSICS measurements on A549 cells co-expressing PA-mEYFP, PB1-mEGFP, and PB2-mCherry2. Data are pooled from four independent experiments. For (**D–F**), the number of cells measured is given in parentheses. Error bars represent mean ± SD.

The online version of this article includes the following source data and figure supplement(s) for figure 6:

**Source data 1.** Relative cross-correlation, normalized molecular brightness values, and diffusion coefficients for three-species raster spectral image correlation spectroscopy measurements on influenza A virus complex and fluorescent protein hetero-oligomers in the nucleus of A549 cells.

**Figure supplement 1.** Cross-correlation and diffusion analysis for three-species raster spectral image correlation spectroscopy (RSICS) measurements on influenza A virus (IAV) polymerase complex as a function of relative protein concentration.

mCherry2, respectively, $p_f$ corrected normalized brightness values of $\varepsilon_{PB1-G} = 2.1 \pm 0.7$ (mean ± SD, n = 53 cells), $\varepsilon_{PA-Y} = 1.8 \pm 0.6$, and $\varepsilon_{PB2-Ch2} = 2.2 \pm 0.7$ were obtained (see Materials and methods for details). These results suggest a 2:2:2 stoichiometry of the IAV PC subunits. Finally, we analyzed the diffusion dynamics of PCs via RSICS (*Figure 6F*). The average D measured for PB1-mEGFP, $D_{PB1-G} = 1.7 \pm 0.6$ μm$^2$/s (mean ± SD, n = 53 cells), was ca. 30% lower than the diffusion coefficients determined for PA-mEYFP- and PB2-mCherry2 ($D_{PA-Y} = 2.5 \pm 0.9$ μm$^2$/s and $D_{PB2-Ch2} = 2.6 \pm 0.7$ μm$^2$/s). This observation is compatible with the above-mentioned presence of a minor fraction of unbound (and thus faster diffusing) PA and PB2 (likely in cells with a lower amount of PB1). A more detailed analysis of the data confirmed this interpretation: the molecular brightness and diffusion coefficient of PA-mEYFP depended on the relative concentration of PB1-mEGFP and PA-mEYFP. Lower brightness and higher diffusion coefficients were obtained in cells where PA-mEYFP was present at much higher concentrations than PB1-mEGFP (*Figure 6—figure supplement 1B,C*).

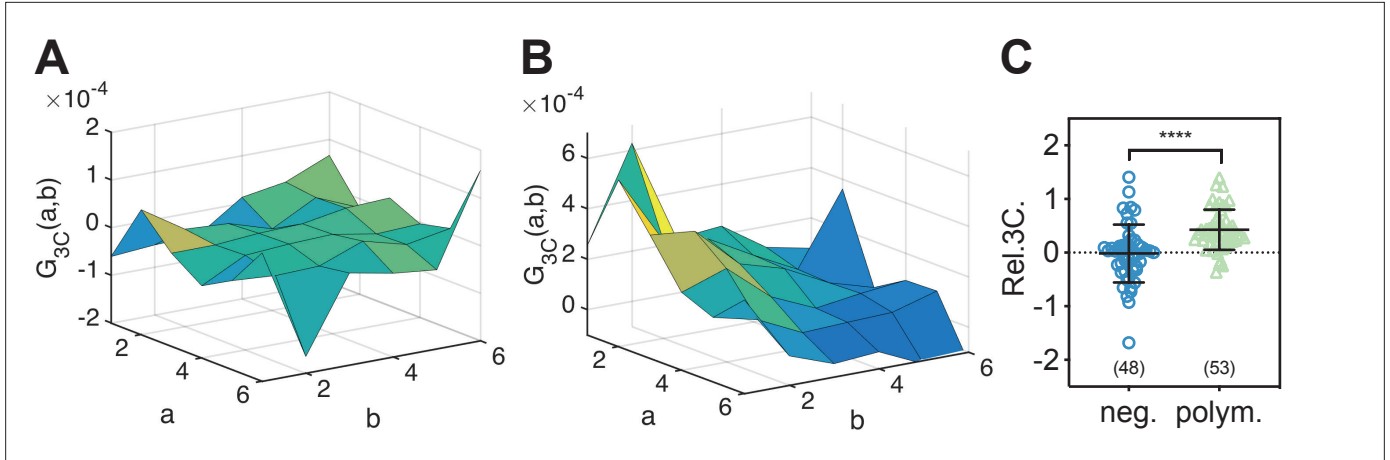

**Figure 7.** Triple raster image correlation spectroscopy (TRICS) reveals the formation of ternary influenza A virus (IAV) polymerase hetero-complexes in the nucleus of A549 cells. (**A, B**) Representative triple-correlation functions (3CFs) obtained from TRICS measurements on A549 cells co-expressing mEGFP, mEYFP, and mCherry2 ('neg.') (**A**) or co-expressing PA-mEYFP, PB1-mEGFP, and PB2-mCherry2 ('polym.') (**B**). The axes *a* and *b* indicate shifts in the x and y direction, respectively, across the three detection channels, as described in Materials and methods. (**C**) Relative triple-correlation (rel.3C.) values obtained from the measurements described in (**A, B**). The number of cells measured is given in parentheses. Error bars represent mean ± SD. Statistical significance was determined using Welch's corrected two-tailed Student's *t*-test (****p<0.0001).

The online version of this article includes the following source data for figure 7:

**Source data 1.** Relative triple correlation values for triple raster image correlation spectroscopy analysis of influenza A virus polymerase complex in the nucleus of A549 cells.

## Triple raster image correlation spectroscopy (TRICS) analysis provides direct evidence for assembly of ternary IAV polymerase complexes

To directly confirm that IAV PC subunits form ternary complexes in the cell nucleus, we implemented a triple-correlation analysis (TRICS) to detect coincident fluctuations of the signal emitted by mEGFP-, mEYFP-, and mCherry2-tagged proteins. A similar analysis has previously been presented for three-channel FCS measurements (e.g., fluorescence triple-correlation spectroscopy [*Ridgeway et al., 2012a*], triple-color coincidence analysis [*Heinze et al., 2004*]), but was so far limited to in vitro systems such as purified proteins (*Ridgeway et al., 2012a*) or DNA oligonucleotides (*Heinze et al., 2004*) labeled with organic dyes. We performed TRICS on data obtained on cells co-expressing PC subunits PA-mEYFP, PB1-mEGFP, and PB2-mCherry2 or cells co-expressing free mEGFP, mEYFP, and mCherry as a negative triple-correlation control. To evaluate ternary complex formation, we quantified the relative triple-correlation (rel.3C., see Materials and methods) for both samples from the amplitudes of the ACFs and triple-correlation functions (3CFs). *Figure 7A and B* show representative 3CFs for the negative control and the PC sample, respectively. For the negative control, we obtained rel.3C. values fluctuating around zero (*Figure 7C*), rel.3C. = −0.02 ± 0.54 (mean ± SD, n = 49 cells). In contrast, significantly higher, positive rel.3C. values were obtained for the polymerase samples, rel.3C. = 0.43 ± 0.38 (mean ± SD, n = 53 cells). The detection of ternary complexes is limited by non-fluorescent FPs, that is, only a fraction of ternary complexes present in a sample will emit coincident signals for all three FP species. In addition, imperfect overlap of the detection volumes for each channel will further reduce the fraction of ternary complexes that can be detected by TRICS. We therefore performed an approximate calculation of the expected rel.3C. value for a sample containing 100% ternary complexes assuming a $p_f$ of 0.7 for each FP species and estimating the reduction due to imperfect overlap from the pair-wise rel.cc. values detected on the positive cross-correlation control (see Appendix 1, Section A1.3 for details). For a 2:2:2 stoichiometry, we obtained an estimated rel.3C. of 0.48, that is, only slightly higher than the average value determined experimentally for IAV PCs. Thus, we estimate that around 90% of PC subunits undergo ternary complex formation in the cell nucleus when all subunits are present.

## Discussion

In this work, we combine FFS techniques with spectral detection to perform multicolor studies of protein interactions and dynamics in living cells. In particular, we present SFSCS, a combination of FSCS (*Benda et al., 2014*) and lateral scanning FCS (*Ries and Schwille, 2006*). We show that SFSCS allows cross-talk-free measurements of protein interactions and diffusion dynamics at the PM of cells and demonstrate that it is capable of detecting three or four species simultaneously. Furthermore, we extend RSICS (*Schrimpf et al., 2018*) to investigate four fluorophore species and apply this approach to determine the stoichiometry of higher order protein complexes assembling in the cell nucleus. Notably, the technical approaches can be carried out on a standard confocal microscope, equipped with a spectral photon counting detector system.

In the first part, we present two-species SFSCS using a single excitation wavelength and strongly overlapping fluorophores. Compared to the conventional implementation of FCCS with two excitation lasers and two detectors, two-species SFSCS has substantial advantages, similar to the recently presented sc-FLCCS (*Štefl et al., 2020*). Since it requires a single excitation line and is compatible with spectrally strongly overlapping FPs, it circumvents optical limitations such as imperfect overlap of the observation volumes. This is evident from higher rel.cc. values of 70–80% measured for mEGFP and mEYFP coupled in FP hetero-oligomers compared to 45–60% observed for mEGFP and mCherry2. Rel.cc. values around 70% are to be expected for the examined FP tandems even in the case of single-wavelength excitation, given that the $p_f$ for such fluorophores is indeed around 0.7 (*Foo et al., 2012*; *Dunsing et al., 2018*) (see also SI, paragraph 1). On the other hand, in three- and four species measurements discussed below, FP pairs requiring two excitation wavelengths display the typical reduction of the rel.cc. due to imperfect optical volume overlap. For combinations of green and red FPs, rel.cc. values below 60% were also observed with single-wavelength excitation (*Foo et al., 2012*; *Shi et al., 2009*), indicating that overlap of both excitation and detection volumes (the latter requiring FPs with similar emission spectra) is required to maximize the achievable cross-correlation (*Foo et al., 2012*). Notably, two-species SFSCS can not only successfully discriminate between mEGFP

and mEYFP, but is also applicable when using the red FPs mApple and mCherry2. These two FPs were successfully used in several FFS studies (*Dunsing et al., 2018*; *Foust et al., 2019*; *Sankaran et al., 2021*), providing the best compromise between brightness, maturation, and photostability among available red FPs, which generally suffer from reduced SNR compared to FPs emitting in the green or yellow part of the optical spectrum (*Dunsing et al., 2017*; *Dunsing et al., 2018*; *Cranfill et al., 2016*).

In comparison to sc-FLCCS, it may be more robust to discriminate fluorophores based on spectra rather than lifetimes, which can be strongly affected by FRET (*Štefl et al., 2020*). The emission spectra of the FPs utilized in this study did not depend on cell lines or subcellular localization (*Figure 5—figure supplement 1*) and showed no (mEGFP, mEYFP) or little (mApple, mCherry2) variation with pH over a range of 5.0–9.2 (*Figure 5—figure supplement 2*). For red FPs, specifically mApple, a red shift appeared at more acidic pH, in agreement with previous studies (*Hendrix et al., 2008*). This aspect should be considered for specific applications, for example, RSICS in the cytoplasm containing acidic compartments such as lysosomes. Generally, spectral approaches require accurate detection of photons in each spectral bin. A previous study using the same detection system reported intrinsic cross-talk between adjacent spectral bins (*Foust et al., 2019*). However, since the methodology presented here is based on temporal (SFSCS) or spatial (RSICS) correlation (both excluding the correlation at zero time or spatial lag), this issue can be neglected in our analysis.

A major limitation of SFSCS is the reduced SNR of the CFs (see *Figure 1*, *Figure 3—figure supplement 3*) caused by the statistical filtering of the signal emitted by spectrally overlapping fluorophore species (see, e.g., *Figure 4—figure supplement 1*). This limitation applies to all FFS methods that discriminate different fluorophore species based on spectral (e.g., FSCS [*Benda et al., 2014*], RSICS [*Schrimpf et al., 2018*]) or lifetime patterns (e.g., sc-FLCCS [*Štefl et al., 2020*]). The increase in noise depends on the spectral (or lifetime) overlap of different species and is more prominent for species that completely lack 'pure' channels, that is, detection channels in which the majority of photons can be univocally assigned to a single species (*Schrimpf et al., 2018*). In sc-FLCCS, this issue particularly compromises the SNR of short lifetime species (*Štefl et al., 2020*) since photons of longer lifetime species are detected in all 'short lifetime' channels at substantial relative numbers. In these conditions, sc-FLCCS could not provide reliable results with sixfold (or higher) difference in relative protein abundance, even though the lower abundant protein was tagged with the brighter, longer lifetime FP (*Štefl et al., 2020*). Similarly in SFSCS, CFs corresponding to mEYFP or mCherry2 were most prone to noise (*Figure 1C and F*) since all channels that contain, for example, mEYFP signal also contain mEGFP signal (*Figure 1—figure supplement 1*). In our experiments, cross-talk-free SFSCS analysis with two species excited with a single excitation wavelength could be performed for relative intensity levels as low as 1:10 (mEGFP/mEYFP) or 1:5 (mApple/mCherry2). In this range, SFSCS not only enabled the quantification of protein interactions via cross-correlation analysis, but also yielded correct estimates of protein diffusion dynamics and oligomerization at the PM. An improvement of the allowed relative concentration range can be achieved by using brighter or more photostable fluorophores, for example, organic dyes, compensating for reduced SNR due to statistical filtering. Alternatively, FP tags could be selected based on proteins' oligomerization state. For example, monomeric proteins exhibiting low molecular brightness should be tagged with fluorophores that are less prone to noise. It should be noted that the limitation of reduced SNR due to excess signal from another species also applies to conventional dual-color FCCS: bleed-through from green to red channels can be corrected on average, but reduces the SNR in red channels (*Bacia et al., 2012*), unless more sophisticated schemes such as pulsed interleaved excitation (*Müller et al., 2005*; *Hendrix et al., 2013*) are applied.

Having demonstrated that two-species SFSCS is feasible with a single excitation wavelength in the green (mEGFP, mEYFP) or red (mApple, mCherry2) part of the visible spectrum, we finally implemented three- and four-species SFSCS as well as four-species RSICS. These extensions do not further compromise the SNR of CFs detected for mEGFP and mEYFP (see *Figure 3—figure supplement 3A,B*), but may additionally reduce the SNR of CFs corresponding to red FPs (in particular when mEGFP and/or mEYFP concentration is much higher than that of red FPs, *Figure 3—figure supplement 3C*). For this reason, three- and four-species analysis was restricted to cells with relative average intensity levels of 1:5 or less between species with adjacent emission spectra. In this range, the increase in noise due to statistical filtering was moderate, benefitting from the fairly large spectral separation of green/yellow and red emission (*Figure 3—figure supplement 3*). In addition, the higher molecular brightness of mApple (compared to mCherry2) compensated for the larger overlap

of this FP with the tail of mEYFP emission. The excitation power for red FPs was generally limited by the lower photostability of mApple, which could be responsible for consistently lower rel.cc. values of mEGFP or mEYFP with mApple than with mCherry2. Nevertheless, four-species SFSCS and RSICS could successfully resolve different combinations of strongly overlapping FP hetero-oligomers, for example, a mixture of mEGFP-mCherry2 and mEYFP-mApple heterodimers, at the PM or in the cytoplasm of cells. To explore the interaction of four different FP-tagged proteins, four-species FFS may substantially reduce the experimental effort because all pair-wise interactions can be quantified in a single measurement (instead of six separate conventional two-species FCCS measurements). Yet, weak interaction of proteins, that is, a low amount of hetero-complexes compared to a high amount of unbound proteins, may not be detectable due to the large noise of the CCF in this case. The SNR might be further compromised by slow FP maturation or dark FP states, limiting the amount of complexes that simultaneously emit fluorescence of all bound FP species (*Dunsing et al., 2018*). Ultimately, the mentioned limitations currently restrict SFSCS and RSICS to four FP species. The approaches would thus strongly benefit from a multiparametric analysis. For instance, combining spectral and lifetime detection schemes would provide additional contrast for photons detected in the same spectral bin. This improvement could expand the range of detectable relative concentrations or might allow further multiplexing of FFS.

Conventional two-color scanning FCCS has been previously applied to quantify receptor-ligand interactions in living zebrafish embryos (*Ries et al., 2009b*) and CRISPR/Cas9 edited cell lines to study such interactions at endogenous protein level (*Eckert et al., 2020*). SFSCS is thus directly applicable in the complex environment of living multicellular organisms. In this context, spectral information could be further exploited to separate low signal levels of endogenously expressed, fluorescently tagged proteins from autofluorescence background.

As a first biological application of SFSCS, we investigated the interaction of IAV matrix protein M2 with two cellular host factors: the tetraspanin CD9 and the autophagosome protein LC3. We observed strong association of LC3 with M2, and consequent recruitment of LC3 to the PM (*Figure 3—figure supplement 4*), in agreement with previous in vitro and localization studies (*Beale et al., 2014*). Interestingly, molecular brightness analysis reported oligomerization (dimers to tetramers) of M2, but indicated a monomeric state of LC3 at the PM, that is, binding of LC3 to M2 in an apparent stoichiometry of 1:2 to 1:4. However, each M2 monomer provides a binding site for LC3 in the cytoplasmic tail (*Claridge et al., 2020*). A more detailed analysis of our data showed that in the analyzed cells (i.e., cells showing clear membrane recruitment of LC3, *Figure 3—figure supplement 4A*,B), the PM concentration of LC3 was on average only 30% compared to that of M2 (*Figure 3—figure supplement 4C*), although both proteins were expressed in comparable amounts in the sample in general. This suggests that not all potential binding sites in the cytoplasmic tail of M2 may be available to fluorescently tagged LC3, either due to binding of endogenous LC3, other cellular host factors, or steric hindrance. In contrast to the case of LC3, we did not detect significant binding of M2 with the tetraspanin CD9, a protein that was previously shown to be incorporated into IAV virions and supposedly plays a functional role during the infection process (*Shaw et al., 2008*; *Hutchinson, 2014*). Of note, we cannot exclude the possibility that the FP tag at the C-terminus of CD9 might hamper interactions with M2, in the specific case of M2-CD9 interaction being mediated by the C-terminal cytoplasmic tails of the two proteins. In future studies, the approach presented here may be used to further elucidate the complex interaction network of viral proteins, for example, matrix protein 1 (M1) (*Hilsch et al., 2014*), M2, HA, and neuraminidase, cellular host factors, and PM lipids (*Bobone et al., 2017*) during the assembly process of IAV at the PM of living cells (*Rossman and Lamb, 2011*).

Finally, we demonstrated that RSICS allows the quantification of the stoichiometry of higher order molecular complexes, based on molecular brightness analysis for each FP species. As example of an application in a biological context, we determined the stoichiometry of the IAV PC. Our data provide strong evidence for a 2:2:2 stoichiometry of the PC subunits PA, PB1, and PB2, that is, dimerization of heterotrimeric PCs. Such interactions were previously proposed based on experiments in solution using X-ray crystallography and cryo-electron microscopy (*Fan et al., 2019*), co-immunoprecipitation assays (*Jorba et al., 2008*; *Nilsson-Payant et al., 2018*), as well as single-channel brightness analysis of FCCS data (for the PA subunit) (*Huet et al., 2010*). Intermolecular interactions in the PC are hypothesized to be required for the initiation of vRNA synthesis during replication of the viral genome (*Fan et al., 2019*; *Chen et al., 2019*). The results presented here provide the first quantification of

these interactions in living cells and a direct estimate of the stoichiometry of PCs in the cell nucleus. The formation of ternary PC complexes in these samples could be extrapolated from the observed high rel.cc. values for all three pair combinations, indicating very low amounts of unbound PA, PB1, or PB2 and higher order interactions (see Appendix 1, Section A1.1 for additional details). Furthermore, this observation could also be directly confirmed by performing, for the first time in living cells, a triple-correlation analysis (TRICS), indicating the presence of a considerable amount of PA-PB1-PB2 complexes. It is worth noting though that the detection of coincident triple fluctuations is prone to considerable noise and thus still limited to molecular complexes present at low concentration and characterized by high molecular brightness for each fluorophore species (*Ridgeway et al., 2012a*; *Ridgeway et al., 2012b*).

Of note, the RSICS approach presented here provides for the first time simultaneous information on molecular interactions, molecular brightness (and thus stoichiometry), diffusion dynamics, and concentration for all three complex subunits. This specific feature opens the possibility of a more in-depth analysis. For example, it is possible to quantify the relative cross-correlation of two subunits, e.g. PA and PB2, as a function of the relative concentration of the third subunit, for example, PB1 (*Figure 6—figure supplement 1A*). Similarly, molecular brightness and diffusion coefficients can be analyzed as a function of the abundance of each subunit (*Figure 6—figure supplement 1B,C*). With this approach, it is therefore possible to distinguish specific molecular mechanisms, such as inefficient PA-PB2 interactions in the presence of low PB1 concentration or efficient heterotrimer dimerization when all subunits are present at similar concentrations. The employed experimental scheme offers a powerful tool for future studies, exploring, for example, interaction of the PC with cellular host factors or the development of inhibitors that could interfere with the assembly process of the complex, as a promising therapeutic target for antiviral drugs (*Massari et al., 2021*).

## Limitations

We summarize in this section the main instrumental, conceptual, and sample-related limitations and requirements connected to the multicolor FFS approach employed in this work.

### Instrumental limitations

To perform multicolor FFS, a spectral photon counting detector system is required. Alternatively, the same conceptual approach can be implemented based on detection of fluorophore lifetimes rather than emission spectra (*Štefl et al., 2020*). For both approaches, two excitation wavelengths are currently required for three- and four-species detection. As a consequence, the overlap of excitation volumes of the two laser lines might be limited, thus reducing the maximum achievable rel.cc., as previously discussed for standard FCCS (*Foo et al., 2012*). For the instrumentation utilized in the present work, the time resolution for SFSCS was limited to 0.5 ms. However, RSICS can be applied to detect faster dynamics, as demonstrated by experiments on cytoplasmic proteins.

### Conceptual limitations

FFS approaches generally require the proteins of interest to diffuse and thus cannot be applied in the case of immobile or strongly clustered targets (*Ciccotosto et al., 2013*). The statistical filtering of spectrally overlapping FP emission leads to increased noise of CFs. FPs lacking 'pure' channels, for example, mEYFP when co-expressed with mEGFP, are most compromised. As a consequence, the approach provides reliable results only in a certain range of relative protein abundance. For the presented three- and four-species SFSCS and RSICS experiments, relative signals were limited to 1:5 (i.e., range of 1:5 to 5:1). The given ratios characterize the minimum acceptable signal ratio for spectrally neighboring fluorescent species, for the FPs utilized in this work. The set of FPs may be optimized for specific applications. The increase in noise as a result of filtering may prevent detection of weak protein interactions due to the low SNR of CCFs in this case. Furthermore, detection of co-fluctuations of three FP species based on triple correlation is prone to considerable noise and thus limited to detection of molecular complexes present at low concentrations or characterized by high molecular brightness, as discussed previously for in vitro studies (*Ridgeway et al., 2012a*).

## Sample-related limitations

To apply multicolor FFS, multiple FP species (e.g., FP-tagged proteins of interest) have to be expressed in the same cell, in relative amounts compatible with the ranges given above. Since tagging of proteins of interest with FPs is required (or other labels such as organic dyes, if the labeling ratio can be precisely determined), potential hindrance of protein interactions by the tags should be carefully evaluated. Typical measures consist in, for example, testing different positions for the tag in the protein of interest, trying different linkers with varying length and flexibility, using tags with smaller sizes, or bio-orthogonal labeling (*Huang et al., 2014*; *Işbilir et al., 2021*). The emission spectra of most FPs are typically well-defined, but might depend on physicochemical conditions (e.g., mApple showed red-shifted emission at more acidic pH). Differences between calibrated and actual spectra could induce errors in filtering and cause residual cross-talk between different FP species (*Schrimpf et al., 2018*). Therefore, the same optical components (e.g., filters, beam splitters) and experimental conditions (e.g., laser powers, sample media, dishes) should be used to calibrate the spectra. Due to lower photostability and quantum yield, red FPs suffer from reduced SNR and, thus, larger variation of parameter estimates compared to green FPs. This is most evident for mCherry2 in four-species applications. In addition, molecular brightness and cross-correlation analysis are compromised by FP maturation. Slow maturation will lead to an increased fraction of dark states, increasing the noise of CCFs and reducing the dynamic range for brightness analysis of protein oligomers (*Dunsing et al., 2018*; *Foo et al., 2012*). Cross-correlation analysis may be further affected by FRET between different FP species, potentially reducing experimental rel.cc. values (*Foo et al., 2012*). This should be carefully evaluated, for example, by analyzing molecular brightness values relative to monomeric references, for both the proteins of interest and FP-hetero-oligomers used to calibrate the maximum achievable rel.cc. FRET artifacts can be minimized using appropriate linkers, for example, rigid linker peptides, as presented here.

## Conclusions

In summary, we present here three-species and, for the first time, four-species measurements of protein interactions and diffusion dynamics in living cells. This is achieved by combining and extending existing FFS techniques with spectrally resolved detection. The presented approaches provide a powerful toolbox to investigate complex protein interaction networks in living cells and organisms.

# Materials and methods
## Cell culture and sample preparation

Human embryonic kidney (HEK) cells from the 293T line (purchased from ATCC, Manassas, VA; CRL-3216TM) and human epithelial lung cells A549 (ATCC, CCL-185TM) were cultured in Dulbecco's modified Eagle medium (DMEM) with the addition of fetal bovine serum (10%), L-glutamine (2 mM), penicillin (100 U/mL), and streptomycin (100 μg/mL). Mycoplasma contamination tests and morphology tests were performed every 3 months and 2 weeks, respectively. Cells were passaged every 3–5 days, no more than 15 times. All solutions, buffers, and media used for cell culture were purchased from PAN-Biotech (Aidenbach, Germany).

For microscopy experiments, $3 \times 10^5$ (HEK) or $4 \times 10^5$ (A549) cells were seeded in 35 mm #1.5 optical glass-bottom dishes (CellVis, Mountain View, CA) 24 hr before transfection. Cells were transfected 16–24 hr prior to the experiment using between 50 ng and 150 ng plasmid per dish with Turbofect (HEK) or Lipofectamin3000 (A549) according to the manufacturer's instructions (Thermo Fisher Scientific, Waltham, MA). Briefly, plasmids were incubated for 20 min with 3 μl Turbofect diluted in 50 μl serum-free medium, or 15 min with 2 μl P3000 and 2 μl Lipofectamine3000 diluted in 100 μl serum-free medium, and then added dropwise to the cells. For spectral imaging at different pH values, culture medium was exchanged with buffer containing 140 mM NaCl, 2.5 mM KCl, 1.8 mM $CaCl_2$, 1.0 mM $MgCl_2$, and 20 mM HEPES with pH ranging from 5.0 to 9.2.

## Plasmids and cloning

The plasmids encoding FPs linked to a myristoylated and palmitoylated peptide (mp-mEGFP, mp-mEYFP, mp-mCherry2, mp-2x-mEGFP), the full-length IAV A/chicken/FPV/Rostock/1934 hemagglutinin (HA) construct HA-mEGFP, and the plasmids for cytosolic expression of mEGFP, mEYFP,

mCherry2, 2x-mEGFP, 2x-mEYFP, 2x-mCherry2, and mCherry2-mEGFP heterodimers were previously described (*Dunsing et al., 2018*) and are available on Addgene.

For the cloning of all following constructs, standard PCRs with custom-designed primers were performed, followed by digestion with fast digest restriction enzymes and ligation with T4-DNA-Ligase according to the manufacturer's instructions. All enzymes and reagents were purchased from Thermo Fisher Scientific.

To obtain mp-mEGFP-mEYFP, a mp-mEGFP_pcDNA3.1+ vector was first generated by amplifying mp-mEGFP insert from the respective plasmid, and inserting it into pcDNA3.1+ vector (obtained from Thermo Fisher Scientific) by digestion with NheI and AflII. Afterwards, mEYFP was amplified from mp-mEYFP and inserted into mp-mEGFP_pcDNA3.1+ using digestion with AflII and KpnI. To clone mp-mEYFP-(L)-mEGFP (a plasmid encoding for mp-mEYFP-mEGFP heterodimers with a long rigid linker peptide [L] between FPs), a mp-mEYFP-(L)_pcDNA3.1+ construct was first generated by amplifying mp-mEYFP from the respective plasmid with primers encoding for the rigid linker (see *Supplementary file 1a* for linker peptide sequences) and inserting it into pcDNA3.1+ vector by digestion with NheI and AflII. Then, mEGFP was inserted from mEGFP-(L)_pcDNA3.1+ (see below) by digestion with KpnI and BamHI. To generate mp-mEYFP-(L)-mCherry2-(L)-mEGFP, a mp-mEYFP-(L)-mCherry2-(L) construct was first cloned by amplifying mCherry2 from a mCherry2-C1 vector (a gift from Michael Davidson, Addgene plasmid # 54563) and inserting it into mp-mEYFP-(L)_pcDNA3.1+ by digestion with AflII and KpnI. Subsequently, mEGFP was inserted from mEGFP-(L)_pcDNA3.1+ (see below) using KpnI and BamHI restriction. The mp-mEYFP-(L)-mCherry2-(L)-mEGFP-(L)-mApple plasmid was generated by inserting an mEGFP-(L)-mApple cassette into mp-mEYFP-(L)-mCherry2-(L) by digestion with KpnI and EcoRI. The mEGFP-(L)-mApple construct was cloned beforehand by amplifying mApple from PMT-mApple (*Sankaran et al., 2021*) (a kind gift from Thorsten Wohland) and inserting it into mEGFP-(L)_pcDNA3.1+ by digestion with BamHI and EcoRI. The mEGFP-(L)_pcDNA3.1+ plasmid was obtained by amplifying mEGFP from an mEGFP-N1 vector (a gift from Michael Davidson, Addgene plasmid #54767) (using a primer encoding a long rigid linker sequence) and inserting it into a pcDNA3.1+ vector by KpnI and BamHI restriction. The mApple_pcDNA3.1+ plasmid was generated by amplifying mApple from PMT-mApple and inserting it into pcDNA3.1+ vector by digestion with KpnI and BamHI. The mp-mApple plasmid was generated by amplifying mApple from PMT-mApple and inserting it into mp-mCherry2 by digestion with AgeI and BsrGI. To clone mp-mCherry2-(L)-mApple, mp-mCherry2-(L)_pcDNA3.1+ plasmid was first generated by amplifying mp-mCherry2 (using a primer encoding a long rigid linker sequence) and inserting it into pcDNA3.1+ using NheI and KpnI restriction. Afterwards, mApple was amplified from PMT-mApple and inserted into mp-mCherry2-(L)_pcDNA3.1+ by digestion with KpnI and EcoRI. The mp-mCherry2-mEGFP plasmid was cloned by inserting mp from mp-mEGFP into mCherry2-mEGFP using digestion with NheI and AgeI. The plasmids mEYFP-(L)-mApple, mEYFP-(L)-mCherry2-(L)-mEGFP, and mEYFP-(L)-mCherry2-(L)-mEGFP-(L)-mApple were generated by amplifying the respective insert from mp-mEYFP-(L)-mApple, mp-mEYFP-(L)-mCherry2-(L)-mEGFP, or mp-mEYFP-(L)-mCherry2-(L)-mEGFP-(L)-mApple and inserting it into pcDNA3.1+ vector by digestion with NheI and XbaI. The mp-mEYFP-(L)-mApple construct was cloned beforehand by inserting mApple from mEGFP-(L)-mApple into mp-mEYFP-(L)_pcDNA3.1+ using restriction by BamHI and EcoRI.

The CD9-mEGFP plasmid was cloned by amplifying CD9 from pCMV3-CD9 (obtained from SinoBiological #HG11029-UT, encoding human CD9) and inserting into mEGFP-C1 vector using restriction by HindIII and BamHI. The LC3-mEYFP plasmid was generated by inserting mEYFP from mEYFP-C1 vector into pmRFP-LC3 (*Kimura et al., 2007*) (a gift from Tamotsu Yoshimori, Addgene plasmid #21075, encoding rat LC3) using digestion with NheI and BglII. Plasmid M2-mCherry2 (mCherry2 fused to the extracellular terminus of matrix protein 2 from influenza A/chicken/FPV/Rostock/1934) was cloned by inserting mCherry2 from an mCherry2-C1 vector into mEYFP-FPV-M2 (a kind gift from Michael Veit) using restriction by AgeI and BsrGI. Plasmids encoding IAV polymerase subunits PA-mEYFP, PB1-mEGFP, and PB2-mCherry2 (from influenza A/human/WSN/1933) were a kind gift from Andreas Herrmann.

The plasmids GPI-mEYFP and GPI-EGFP were a kind gift from Roland Schwarzer. GPI-mEGFP was cloned by amplifying mEGFP from an mEGFP-N1 vector and inserting it into GPI-EGFP using digestion with AgeI and BsrGI. To generate GPI-mApple and GPI-mCherry2, mApple and mCherry2 inserts

were amplified from PMT-mApple and mCherry2-C1, respectively, and inserted into GPI-mEYFP using restriction by AgeI and BsrGI.

All plasmids generated in this work will be made available on Addgene.

## Confocal microscopy system

SFSCS and RSICS were performed on a Zeiss LSM880 system (Carl Zeiss, Oberkochen, Germany) using a 40× , 1.2 NA water immersion objective. For two-species measurements, samples were excited with a 488 nm argon laser (mEGFP, mEYFP) or a 561 nm diode laser (mCherry2, mApple). For three- and four-species measurements, both laser lines were used. To split excitation and emission light, 488 nm (for two-species measurements with mEGFP and mEYFP) or 488/561 nm (for measurements including mCherry2 and mApple) dichroic mirrors were used. Fluorescence was detected in spectral channels of 8.9 nm (15 channels between 491 nm and 624 nm for two-species measurements on mEGFP, mEYFP; 14 channels between 571 nm and 695 nm for two-species measurements on mCherry2, mApple; 23 channels between 491 nm and 695 nm for three- and four-species measurements) on a 32-channel GaAsP array detector operating in photon counting mode. All measurements were performed at room temperature.

## Scanning fluorescence spectral correlation spectroscopy (SFSCS)

### Data acquisition

For SFSCS measurements, line scans of 256 × 1 pixels (pixel size 80 nm) was performed perpendicular to the PM with 403.20 µs scan time. This time resolution is sufficient to reliably detect the diffusion dynamics observed in the samples described in this work (i.e., diffusion times ~6–60 ms). Typically, 450,000–600,000 lines were acquired (total scan time ca. 2.5–4 min). Laser powers were adjusted to keep photobleaching below 50% at maximum for all species (average signal decays were ca. 10% for mEGFP, 30% for mEYFP, 40% for mApple, and 20% for mCherry2). Typical excitation powers were ca. 5.6 µW (488 nm) and ca. 5.9 µW (561 nm). Spectral scanning data were exported as TIFF files (one file per three spectral channels), imported, and analyzed in MATLAB (The MathWorks, Natick, MA) using custom-written code (*Dunsing and Chiantia, 2021*).

### Data analysis

SFSCS analysis followed the scanning FCS scheme described previously (*Ries and Schwille, 2006*; *Dunsing and Chiantia, 2018*), combined with spectral decomposition of the fluorescence signal by applying the mathematical framework of FLCS and FSCS (*Benda et al., 2014*; *Böhmer et al., 2002*). Briefly, all scan lines were aligned as kymographs and divided in blocks of 1000 lines. In each block, lines were summed up column-wise and across all spectral channels, and the lateral position with maximum fluorescence was determined. This position defines the membrane position in each block and was used to align all lines to a common origin. Then, all aligned line scans were averaged over time and fitted with a Gaussian function. The pixels corresponding to the PM were defined as pixels within ±2.5 SD of the peak. In each line and spectral channel, these pixels were integrated, providing membrane fluorescence time series $F^k(t)$ in each spectral channel k (m channels in total). These time series were then temporally binned with a binning factor of 2 and subsequently transformed into the contributions $F_i(t)$ of each fluorophore species i (i.e., one fluorescence time series for each species) by applying the spectral filtering algorithm presented by *Benda et al., 2014*:

$$F_i(t) = \sum_{k=1}^m f_i^k F^k(t).$$

Spectral filter functions $f_i^k$ were calculated based on reference emission spectra $p_i^k$ that were determined for each individual species i from single species measurements performed on each day using the same acquisition settings:

$$f_i^k = \left( \left[ \hat{M}^T D \hat{M} \right]^{-1} \hat{M} D \right)_{ik}.$$

Here, $\hat{M}$ is a matrix with elements $M_{ki} = p_i^k$ and D is a diagonal matrix, $D = diag\left[ 1/\left\langle F^k(t) \right\rangle \right]$.

In order to correct for depletion due to photobleaching, a two-component exponential function was fitted to the fluorescence time series for each spectral species, $F_i(t)$, and a correction formula

was applied (***Dunsing and Chiantia, 2018***; ***Ries et al., 2009a***). Finally, ACFs and pair-wise CCFs of fluorescence time series of species i and j were calculated as follows using a multiple tau algorithm:

$$G_{i,j}\left(\tau\right) = \frac{\langle \delta F_i(t)\delta F_j(t+\tau)\rangle}{\langle F_i(t)\rangle\langle F_j(t)\rangle},$$

where $\delta F_i\left(t\right) = F_i\left(t\right) - \langle F_i\left(t\right)\rangle$.

To avoid artifacts caused by long-term instabilities or single bright events, CFs were calculated segment-wise (10–20 segments) and then averaged. Segments showing clear distortions (typically less than 25% of all segments) were manually removed from the analysis (***Dunsing and Chiantia, 2018***).

A model for two-dimensional diffusion in the membrane and Gaussian focal volume geometry (***Ries and Schwille, 2006***) was fitted to all CFs:

$$G\left(\tau\right) = \frac{1}{N}\left(1 + \frac{\tau}{\tau_d}\right)^{-1/2}\left(1 + \frac{\tau}{\tau_d S^2}\right)^{-1/2}$$

To ensure convergence of the fit for all samples (i.e., ACFs and CCFs of correlated and uncorrelated data), positive initial fit values for the particle number $N$ and thus $G\left(\tau\right)$ were used. In the case of uncorrelated data, that is, for CFs fluctuating around zero, this constraint can generate low, but positive correlation amplitudes due to noise. This issue can be circumvented, if needed, by selecting adaptive initial values, for example, obtaining the initial amplitude value from averaging the first points of the CFs (see ***Figure 3—figure supplement 2***).

To calibrate the focal volume, point FCS measurements with Alexa Fluor 488 (Thermo Fisher Scientific) dissolved in water at 20 nM were performed at the same laser power. The structure parameter $S$ was fixed to the average value determined in calibration measurements (typically between 4 and 8).

From the amplitudes of ACFs and CCFs, rel.cc. values were calculated for all cross-correlation combinations:

$$rel.cc._{i,j} = max\left\{\frac{G_{i,j}(0)}{G_i(0)}, \frac{G_{i,j}(0)}{G_j(0)}\right\},$$

where $G_{i,j}(0)$ is the amplitude of the CCF of species $i$ and $j$, and $G_i(0)$ the amplitude of the ACF of species $i$. The molecular brightness was calculated by dividing the mean count rate detected for each species $i$ by the particle number $N_i$ determined from the fit: $B_i = \frac{\langle F_i(t)\rangle}{N_i}$. From this value, an estimate of the oligomeric state $\varepsilon_i$ was determined by normalizing $B_i$ by the average molecular brightness $B_{i,1}$ of the corresponding monomeric reference, and, subsequently, by the fluorescence probability $p_{f,i}$ for species $i$: $\varepsilon_i = \frac{\frac{B_i}{B_{i,1}}-1}{p_{f,i}} + 1$, as previously derived (***Dunsing et al., 2018***). The $p_f$ was previously characterized for several FPs (for example, ca. 60% for mCherry2) (***Dunsing et al., 2018***).

The SNR of the ACFs was calculated by dividing ACF values by their variance and summing over all points of the ACF. The variance of each point of the ACF was calculated in the multiple tau algorithm (***Wohland et al., 2001***).

To ensure statistical robustness of the SFSCS analysis and sufficient SNR, the analysis was restricted to cells expressing all fluorophore species in comparable amounts, that is, relative average signal intensities of less than 1:10 (mEGFP/mEYFP) or 1:5 (mApple/mCherry2, three- and four-species measurements).

## Raster spectral image correlation spectroscopy (RSICS)

### Data acquisition

RSICS measurements were performed as previously described (***Ziegler et al., 2020***). Briefly, 200–400 frames of 256 × 256 pixels were acquired with 50 nm pixel size (i.e., a scan area of 12.83 × 12.83 µm$^2$ through the midplane of cells), 2.05 µs or 4.10 µs pixel dwell time, 1.23 ms or 2.46 ms line, and 314.57 ms or 629.14 ms frame time (corresponding to ca. 2 min total acquisition time per measurement). Samples were excited at ca. 5.6 µW (488 nm) and 4.6 µW (561 nm) excitation powers, respectively. Laser powers were chosen to maximize the signal emitted by each fluorophore species but keeping photobleaching below 50% at maximum for all species (average signal decays were ca. 10% for mEGFP, 15% for mEYFP, 40% for mApple, and 25% for mCherry2). Typical counts per molecule were ca. 25 kHz for mEGFP (G), 15–20 kHz for mEYFP (Y), 20–30 kHz for mApple (A), and 5–10 kHz

for mCherry2 (Ch2). To obtain reference emission spectra for each individual fluorophore species, four image stacks of 25 frames were acquired at the same imaging settings on single-species samples on each day.

## Data analysis

RSICS analysis followed the implementation introduced recently (*Schrimpf et al., 2018*), which is based on applying the mathematical framework of FLCS and FSCS (*Benda et al., 2014*; *Böhmer et al., 2002*) to RICS. Four-dimensional image stacks $I(x, y, t, k)$ (time-lapse images acquired in $k$ spectral channels) were imported in MATLAB (The MathWorks) from CZI image files using the Bioformats package (*Linkert et al., 2010*) and further analyzed using custom-written code (*Dunsing and Chiantia, 2021*). First, average reference emission spectra were calculated for each individual fluorophore species from single-species measurements. Four-dimensional image stacks were then decomposed into three-dimensional image stacks $I_i(x, y, t)$ for each species $i$ using the spectral filtering algorithm presented by *Schrimpf et al., 2018* (following the mathematical framework given in the SFSCS section). Cross-correlation RICS analysis was performed in the arbitrary region RICS framework (*Hendrix et al., 2016*). To this aim, a polygonal region of interest (ROI) was selected in the time- and channel-averaged image frame containing a homogeneous region in the cytoplasm (four-species measurements on FP constructs) or nucleus (three-species measurements on polymerase complex and related controls) of cells. This approach allowed excluding visible intracellular organelles or pixels in the extracellular space, but to include all pixels containing signal from the nucleus of cells. In some cells, nucleus and cytoplasm could not be clearly distinguished. In these cases, all pixels were selected and minor brightness differences between cytoplasm and nucleus, previously found to be ca. 10% (*Dunsing et al., 2018*), were neglected. Image stacks were further processed with a high-pass filter (with a moving four-frame window) to remove slow signal variations and spatial inhomogeneities. Afterwards, RICS spatial ACFs and pair-wise CCFs were calculated for each image stack and all combinations of species $i, j$ (e.g., G and Y, G and Ch2, Y and Ch2 for three species), respectively (*Schrimpf et al., 2018*; *Hendrix et al., 2016*):

$$G_i(\xi, \psi) = \frac{\langle \delta I_i(x,y)\, \delta I_i(x+\xi, y+\psi)\rangle}{\langle I_i(x,y)\rangle^2},$$

$$G_{i,j}(\xi, \psi) = \frac{\langle \delta I_i(x,y)\, \delta I_j(x+\xi, y+\psi)\rangle}{\langle I_i(x,y)\rangle \langle I_j(x,y)\rangle},$$

where $\delta I_i(x, y) = I_i(x, y) - \langle I_i(x, y)\rangle$.

ACF amplitudes were corrected as described in *Hendrix et al., 2016* to account for the effect of the high-pass filter. A three-dimensional normal diffusion RICS fit model (*Digman et al., 2005*; *Digman et al., 2009b*) for Gaussian focal volume geometry (with particle number $N$, diffusion coefficient $D$, waist $\omega_0$, and structure parameter $S$ as free fit parameters) was then fitted to both ACFs and CCFs:

$$G(\xi, \psi) = \frac{1}{N}\left(1 + \frac{4D\left|(\xi - \xi_0)\tau_p + \psi \tau_l\right|}{\omega_0^2}\right)^{-1}\left(1 + \frac{4D\left|(\xi - \xi_0)\tau_p + \psi \tau_l\right|}{\omega_0^2 S^2}\right)\, exp\left(-\frac{\delta s^2\left((\xi - \xi_0)^2 + \psi^2\right)}{\omega_0^2 + 4D\left|(\xi - \xi_0)\tau_p + \psi \tau_l\right|}\right),$$

where $\tau_p$, $\tau_l$ denote the pixel dwell and line time and $\delta s$ the pixel size. The free parameter $\xi_0$ (starting value = 13 pixels) was used to determine which CCFs were too noisy (i.e., $\xi_0 > 4$ pixels) to obtain meaningful parameters (typically in the absence of interaction). For ACF analysis, $\xi_0$ was set to 0. To remove shot noise contributions, the correlation at zero lag time was omitted from the analysis.

From the fit amplitudes of the ACFs and CCFs, rel.cc. values were calculated:

$$rel.cc._{i,j} = max\left\{\frac{G_{i,j}(0,0)}{G_i(0,0)}, \frac{G_{i,j}(0,0)}{G_j(0,0)}\right\},$$

where $G_{i,j}(0,0)$ is the amplitude of the CCF of species $i$ and $j$, and $G_i(0,0)$ the ACF amplitude of species $i$. In the case of non-meaningful convergence of the fit to the CCFs (i.e., $\xi_0 > 4$ pixels), the rel.cc. was simply set to 0. To ensure statistical robustness of the RSICS analysis and sufficient SNR, the analysis was restricted to cells expressing all fluorophore species in comparable amounts, that is, relative average signal intensities of less than 1:6 for all species (in all RSICS experiments). The molecular brightness of species $i$ was calculated by dividing the average count rate in the ROI by the particle

number determined from the fit to the ACF: $B_i = \frac{\langle I_i(t) \rangle}{N_i}$. From this value, an estimate of the oligomeric state $\varepsilon_i$ was determined by normalizing $B_i$ by the average molecular brightness $B_{i,1}$ of the corresponding monomeric reference, and, subsequently, by the fluorescence probability $p_{f,i}$ for species $i$: $\varepsilon_i = \frac{\frac{B_i}{B_{i,1}} - 1}{p_{f,i}} + 1$, as previously derived (**Dunsing et al., 2018**). The $p_f$ was calculated from the obtained molecular brightness $B_{i,2}$ of FP homodimers of species $i$: $p_f = \frac{B_{i,2}}{B_{i,1}} - 1$ (**Dunsing et al., 2018**).

## TRICS analysis

TRICS was performed using three-dimensional RSICS image stacks $I_i(x, y, t)$ detected for three species $i$. First, the spatial 3CF was calculated:

where $\xi_1, \xi_2$ denote spatial lags along lines and $\psi_1, \psi_2$ along columns of the image stacks. Contributions from $\delta I$ triplets containing at least two intensity values from the same pixel position were not included in the calculation in order to avoid shot-noise artifacts (since all channels are detected here by the same detector). From the resulting four-dimensional matrix, a two-dimensional representation was calculated by introducing coordinates $a$, $b$ for the effective spatial shift between signal fluctuations evaluated for the two-species combinations:

$$a = ceil\left( \sqrt{\xi_1^2 + \xi_2^2} \right),$$

$$b = ceil\left( \sqrt{\psi_1^2 + \psi_2^2} \right).$$

The four-dimensional triple-correlation matrix was transformed into a two-dimensional representation $G_{3C}(a,b)$ by rounding up $a$ and $b$ to integer values and averaging all points with the same rounded spatial shift. For example, for a one-pixel shift along a line in one FP channel and a one-pixel shift along a column in the third FP channel (i.e., $\xi_1 = 1, \psi_1 = 0, \xi_2 = 0, \psi_2 = 1$), $a = b = 1$. $G_{3C}(1,1)$ also includes in its averaged value the other seven correlation values corresponding, for example, to $(\xi_1 = 0, \psi_1 = 1, \xi_2 = 1, \psi_2 = 0)$, $(\xi_1 = 1, \psi_1 = 0, \xi_2 = 0, \psi_2 = -1)$, etc. As a further example, $G_{3C}(2,0)$ includes and averages only the two correlation values corresponding to $\psi_1 = \psi_2 = 0$ (i.e., no shift along columns) and $\xi_1 = -\xi_2 = \pm 1$ (i.e., a one-pixel shift along a line, in opposite directions for the two channels). Note that the combinations $(\psi_1 = \psi_2 = 0, \xi_1 = \pm 2, \xi_2 = 0)$ and $(c\psi_1 = \psi_2 = 0, \xi)$ would also result in $a = 2$ and $b = 0$, but these values were not included since they refer to a correlation between identical pixel positions (e.g., $\xi_2 = 0, \psi_2 = 0$) between two FP channels and would be influenced by shot-noise artifacts (see above).

To determine the triple-correlation amplitude $G_{3C}(0,0)$, the closest points (e.g., $G_{3C}(1,1)$, $G_{3C}(1,2)$, $G_{3C}(2,1)$, $G_{3C}(2,2)$, $G_{3C}(3,0)$) of the two-dimensional triple correlation were averaged as an (slightly underestimated) approximation of the amplitude value at (0,0). Note that we chose not to include $G_{3C}(2,0)$ because this point is the average of only two possible spatial shift combinations, resulting in large statistical noise. Also, the point $G_{3C}(0,3)$ was not included since it refers to shifts along columns (i.e., the slow scanning direction), which, in turn, are characterized by a steeper decrease in amplitude. Finally, for best visualization, $G_{3C}$ is plotted for $a$ and $b$ values $\geq 1$ (see **Figure 7** and **Appendix 1—figure 2**).

To account for reduction of the triple-correlation amplitude due to the high-pass filter, an empirical correction was applied based on simulated triple-correlation amplitudes with different sizes $F$ of the moving window (see Appendix 1, Section A1.2 and **Appendix 1—figure 1**). Notably, applying this empirical correction to the auto- and cross-correlation amplitudes confirmed the previously introduced correction formula (see **Appendix 1—figure 1**), $G_{corr}(\xi, \psi) = \frac{F}{F-1} G(\xi, \psi)$ (**Hendrix et al., 2016**). The triple-correlation amplitude is related to the number of triple complexes $N_{3C}$ (**Heinze et al., 2004**; **Palmer and Thompson, 1987**):

where $N_i$ is the total number of proteins detected for species $i$. In analogy to the rel.cc., a relative triple correlation rel.3C. is defined, quantifying the fraction of triple complexes relative to the total number of proteins of the species that is present in the lowest concentration:

$$rel.3C. = max\left\{ \tfrac{N_{3C}}{N_1}, \tfrac{N_{3C}}{N_2}, \tfrac{N_{3C}}{N_3} \right\} = \tfrac{3}{4} \frac{G_{3C}(0,0)}{G_1(0,0) G_2(0,0) G_3(0,0)} max\left\{ G_1(0,0), G_2(0,0), G_3(0,0) \right\}$$

## Statistical analyses

All data are displayed as scatter dot plots indicating mean values and SDs. Sample size is given in parentheses in each graph. Statistical significance was tested using Welch's corrected two-tailed Student's t-test in GraphPad Prism 7.0 (GraphPad Software) and p-values are given in figure captions.

## Acknowledgements

This work was financed by the German Research Foundation (DFG) grant 254850309 to SC. The LSM 880 instrumentation was funded by the DFG grant INST 336/114-1 FUGG. We kindly thank Madlen Luckner for providing the plasmids for PA-mEYFP, PB1-mEGFP, and PB2-mCherry2 expression, Thorsten Wohland for providing the PMT-mApple plasmid, and Jelle Hendrix for fruitful discussion.

## Additional information

### Funding

| Funder | Grant reference number | Author |
|---|---|---|
| Deutsche Forschungsgemeinschaft | 254850309 | Salvatore Chiantia |

The funders had no role in study design, data collection and interpretation, or the decision to submit the work for publication.

### Author contributions

Valentin Dunsing, Conceptualization, Data curation, Formal analysis, Investigation, Methodology, Project administration, Resources, Software, Visualization, Writing - original draft, Writing – review and editing; Annett Petrich, Investigation, Resources, Writing – review and editing; Salvatore Chiantia, Conceptualization, Formal analysis, Funding acquisition, Methodology, Project administration, Resources, Software, Supervision, Visualization, Writing - original draft, Writing – review and editing

### Author ORCIDs

Valentin Dunsing http://orcid.org/0000-0003-2482-1498
Annett Petrich http://orcid.org/0000-0002-2372-826X
Salvatore Chiantia http://orcid.org/0000-0003-0791-967X

### Decision letter and Author response

Decision letter https://doi.org/10.7554/eLife.69687.sa1
Author response https://doi.org/10.7554/eLife.69687.sa2

## Additional files

### Supplementary files

• Supplementary file 1. Linker sequences for fluorescent protein (FP) hetero-oligomer constructs and day-to-day variability of molecular brightness values obtained three-species raster spectral image correlation spectrosopcy (RSICS). (a): Linker sequences of FP hetero-oligomer constructs. (b): Day-to-day variability of molecular brightness values obtained from three-species RSICS measurements.

• Transparent reporting form

### Data availability

All data generated or analysed during this study are included in the manuscript and supporting files. Source data files have been provided for Figures 1-7. The analysis software is freely available on GitHub: https://github.com/VaDu8989/SpectralFFS.

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

## Appendix 1

### A1.1 Is pair-wise cross-correlation analysis sufficient to detect ternary interactions?

Generally, pair-wise cross-correlation analysis can only detect pair-wise interactions between fluorescently tagged protein species. To understand whether this analysis is sufficient to indicate the presence of heterotrimeric protein complexes for the specific case reported in this work, we investigated brightness and rel.cc. data obtained by RSICS measurements of IAV PC proteins in more detail.

For all three protein species (PA-mEYFP, PB1-mEGFP, PB2-mCherry2, referred here simply as A, B, and C), normalized brightness values close to the values of FP-homodimers were observed in this work. As a simple approximation, we assume therefore that each species, independently of its participation in hetero-complexes, is either (i) exclusively dimeric or (ii) present as a well-defined mixture of monomers and homotrimers. For the latter case, the fraction of monomers ($f_{1,i}$) and trimers ($f_{3,i}$) for each species $i$ can be calculated from the average molecular brightness $\langle \varepsilon \rangle_i$ :

$$f_{1,i} = \frac{1}{1 + \frac{\varepsilon_{1,i}\left(\varepsilon_{1,i} - \langle \varepsilon \rangle_i\right)}{\varepsilon_{3,i}\left(\langle \varepsilon \rangle_i - \varepsilon_{3,i}\right)}},$$

$$f_{3,i} = \frac{1}{1 + \frac{\varepsilon_{3,i}\left(\langle \varepsilon \rangle_i - \varepsilon_{3,i}\right)}{\varepsilon_{1,i}\left(\varepsilon_{1,i} - \langle \varepsilon \rangle_i\right)}},$$

where $\varepsilon_{1,i}$ and $\varepsilon_{3,i}$ denote the molecular brightness of monomers and trimers, respectively.

We then calculate the maximum rel.cc. amplitudes that can be expected in the presence of optimal pair-wise interactions, while still assuming a negligible concentration of complexes containing A, B, and C.

Generally, the ACF and CCF amplitudes for multiple populations (i.e., complexes of species $i$ and $j$ with variable stoichiometry) are calculated as follows (*Kim et al., 2005*):

where $\varepsilon_{k,i}$ and $\varepsilon_{k,j}$ denote the molecular brightness of species $i$ and $j$ (assumed here to be the same for all species) for population $k$ , present at a concentration $C_k$ in the effective volume $V_{eff}$.

For the sake of simplicity, we discuss here only two simple possible scenarios for the two mixtures discussed above (i.e., each PC protein being present exclusively as homodimers or as a mixture of monomers and homotrimers), in the absence of complexes containing all three PC subunits:

1. Homodimers interacting with homodimers of the other species (i.e., AA-BB, AA-CC, BB-CC).
2. Monomers and oligomers interacting (exclusively) with monomers or oligomers of the other species (i.e., A-B, A-C, B-C, AAA-BBB, AAA-CCC, BBB-CCC).

The two scenarios evaluated here correspond to configurations with the highest possible pair-wise correlations (in the absence of complexes containing A, B, and C), still compatible with an average oligomerization value of 2.

For the two scenarios, we calculate ACF and CCF amplitudes according to the formulas given above, assuming the same total concentration for all species and replacing the concentrations by the derived relative fractions of monomers and oligomers. For each scenario, we determine rel.cc. values from the ratio of CCF and ACF amplitudes. Finally, we extend our calculations by considering incomplete maturation of FP tags based on the fluorescence probability $P_f$ For simplicity, we assume the same $P_f$ for each FP species, in agreement with the similar $P_f$ values of ca. 60–75% observed here for mEGFP, mEYFP, and mCherry2. We use a binomial model for the relative occurrence of different subpopulations in each species (*Dunsing et al., 2018*). For example, actual trimers give rise to a fraction $f_k$ of fluorescent trimers ($k = 3$), dimers ($k = 2$), or monomers (k = 1) with a relative occupancy of $f_k = \binom{3}{k} p_f^k \left(1 - p_f\right)^{3-k}$ and brightness $k\varepsilon_1$.

The obtained rel.cc. values for all models are given in *Appendix 1—table 1* for = 1 or $P_f$ = 0.7. For comparison, we also calculated rel.cc. values of the positive control, that is, the maximum pair-

wise rel.cc. for 1:1 stoichiometry heterodimers (A-B/A-C/B-C) or 1:1:1 stoichiometry heterotrimers (A-B-C), resulting in values of 1 (for $P_f$ = 1) and 0.7 (for $P_f$ = 0.7). Experimentally, this control would also account for suboptimal overlap of the detection volumes for each FP combination, which we neglected here for simplicity. In the absence of ternary hetero-interactions, the determined rel.cc. values are at maximum 59% of the rel.cc. of the positive control (i.e., 0.59 for $P_f$ = 0.7 for scenario 1). Higher normalized values (up to 1.19, see *Appendix 1—table 1*) can be obtained only in the presence of hetero-complexes involving all three PC subunits, which we calculated for comparison for the two mixtures (i.e., AA-BB-CC, or A-B-C in mixtures with AAA-BBB-CCC) and both $P_f$ values.

**Appendix 1—table 1.** Relative cross-correlation (rel.cc.) values (here, same for all channel combinations) for pair-wise or ternary interactions of three-species mixtures.
Values in brackets for $p_f$ = 0.7 give rel.cc. values normalized to that of the positive control (i.e., the pair-wise rel.cc. for 1:1 stoichiometry).

| Binding model | $p_f$ = 1 | $p_f$ = 0.7 |
| --- | --- | --- |
| Pair-wise interactions of dimers (e.g., AA-BB, AA-CC, BB-CC) | 0.50 | 0.41 (0.59) |
| Pair-wise interactions of monomers and homotrimers (e.g., A-B, A-C, B-C, AAA-BBB, AAA-CCC, BBB-CCC) | 0.5 | 0.40 (0.57) |
| Positive control (A-B/A-C/B-C or A-B-C) | 1.0 | 0.7 (1.0) |
| Ternary interactions of dimers (e.g., AA-BB-CC) | 1.0 | 0.83 (1.19) |
| Ternary interactions of monomers and trimers (e.g., A-B-C, AAA-BBB-CCC) | 1.0 | 0.80 (1.14) |

Of note, in our experiments, rel.cc.values > 0.7 (relative to the positive control) were observed for all pair-wise interactions between PC subunits (detected average pair-wise rel.cc. values normalized to the positive control were 0.71 for B-C, 0.97 for A-C, and 1.43 for A-B, see *Figure 6D*). As shown based on the different binding models, such high pair-wise rel.cc. values are only possible if ternary complexes are present. Thus, by combining molecular brightness and cross-correlation analysis, we conclude that PC proteins form a substantial amount of ternary complexes in the nucleus of cells.

## A1.2 TRICS analysis of simulated three-species RICS data

To evaluate the performance of TRICS, we first analyzed simulated RICS data. We ran Monte Carlo simulations of three-species RICS for either (i) three independently diffusing species A, B, C or (ii) a heterotrimeric species (e.g., A-B-C complexes). Two-dimensional diffusion and image acquisition was simulated with the following parameters: diffusion coefficient $D$ = 1 μm²/s (set to be the same for all species), N = 1000 particles (for each species), waist $\omega_0$ =0.2 μm, pixel size $\delta s$=0.05 μm, pixel dwell time $\tau_p$ =2 μs, 256 × 256 pixels, 100 frames. RICS ACFs, CCFs, and the TRICS 3CF were calculated. To correct for the reduction of the triple correlation due to the high-pass filter (with filter size of $F$ frames), an empirical correction was applied. To this aim, the variance and third central moment of a series of $10^5$ random numbers, sampled from a Poissonian distribution (with mean $f_0$ = 10), were calculated within windows with variable size $\Delta F$ (*Appendix 1—figure 1*). The empirical function $f_i\left(\Delta F\right) = f_0 \left(\frac{\Delta F - 1}{\Delta F}\right)^{b_i}$ was fitted to the variance ($i$ = 2) and third central moment ($i$ = 3). For the variance and third central moment, $b_2$ = 1.0 and $b_3$ = 3.4 were obtained, respectively. Thus, the reduction of variance and third central moment for a given value $F$ can be corrected using the factor $\left(\frac{\Delta F}{\Delta F - 1}\right)^{b_i}$. For the variance, the determined value $b_2$ is in agreement with a previously discussed correction (*Hendrix et al., 2016*), which was used here to correct experimental ACFs and CCFs. To test whether 3CFs can be effectively corrected with the obtained $\left(\frac{\Delta F}{\Delta F - 1}\right)^{b_3}$ factor, 3CFs were calculated with variable $F$ (in the range 2–16) and the amplitude values determined without or with the correction. In the latter case, fairly constant 3CF amplitudes were obtained, agreeing with the 3CF amplitude calculated without the high-pass filter (data not shown). Exemplary 3CFs for the two simulated scenarios are shown in *Appendix 1—figure 2*. As expected, the rel.3C. values are close to 100 % in the case of heterotrimers and 0% in the case of independently diffusing monomers. The slight underestimation of the rel.3C. for heterotrimers is likely due to the approximated interpolation of the amplitude value from only the first five points of the 3CF.

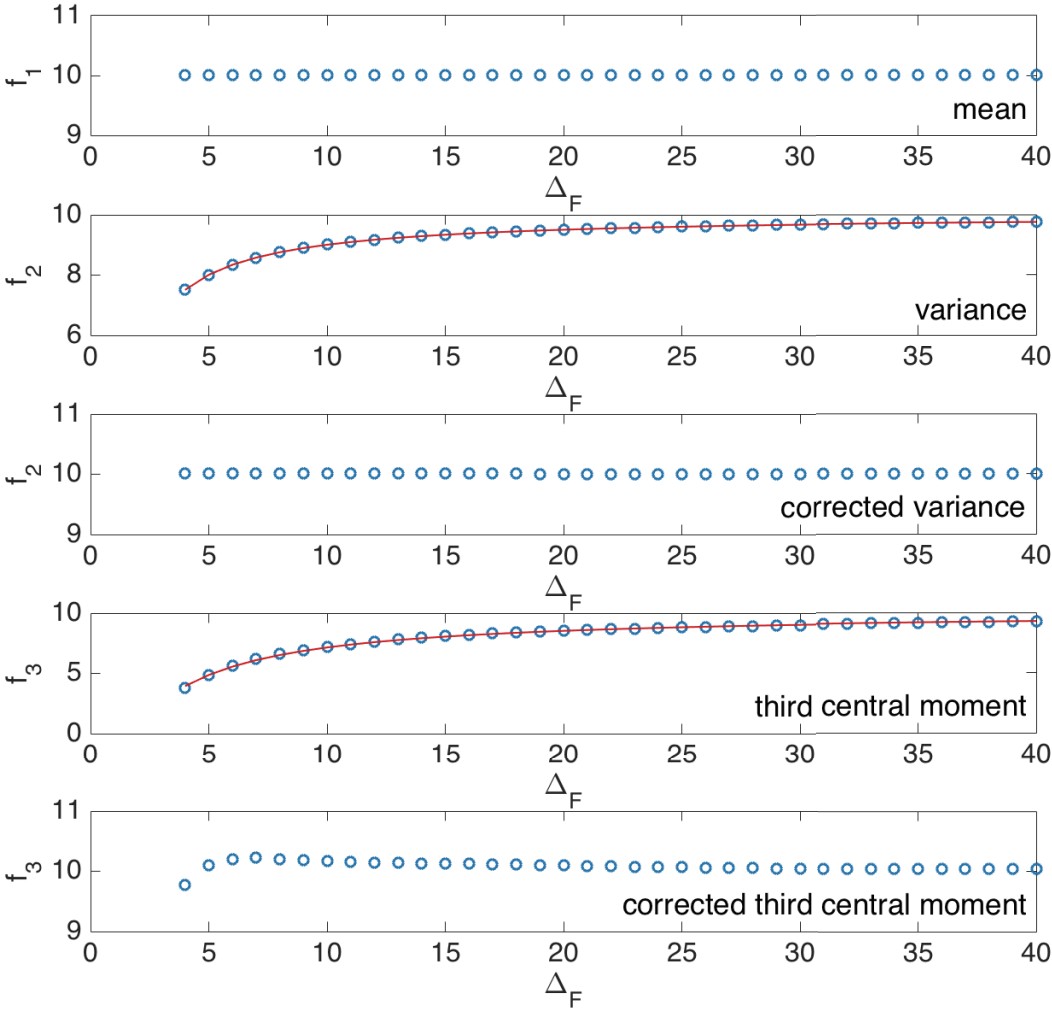

**Appendix 1—figure 1.** Effect of high-pass filter on calculation of variance and third central moment of random numbers sampled from a Poissonian probability distribution. Variance ($f_2$, blue circles) and third central moment ($f_3$, blue circles) were calculated with a moving average (window size $\Delta F$) for a set of $10^5$ random numbers drawn from a Poissonian distribution with average 10. An empirical function (red solid line) of the form $f_i\left(\Delta F\right) = f_0 \left(\frac{\Delta F - 1}{\Delta F}\right)^{b_i}$ was fitted to the variance ($f_2$) and third central moment ($f_3$), and used to correct for the undersampling effect. The corresponding values after applying the empirical correction are shown as blue circles in the panels labeled as 'corrected.'

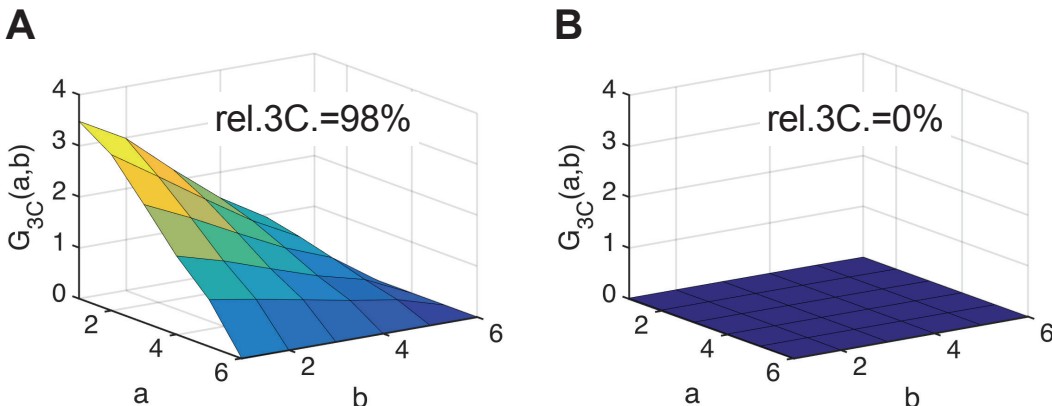

**Appendix 1—figure 2.** Triple raster image correlation spectroscopy (TRICS) analysis of simulated three-species RICS data. (**A, B**) Two-dimensional representation of the triple-correlation function (3CF) calculated for simulated TRICS data (with a four-frame high-pass filter) for (**A**) ternary hetero-complexes or (**B**) the same number of particles per species diffusing as independent monomers. From a linear interpolation of $G_{3C}$ to (0,0) (using the first point $G_{3C}(1,1)$ and the average of the four points $G_{3C}(1,2)$, $G_{3C}(2,1)$, $G_{3C}(2,2)$, $G_{3C}(3,0)$) an approximate value of the 3CF amplitude was determined and corrected with the correction factor discussed in Section A1.1. The obtained value and the autocorrelation function (ACF) amplitude value (also corrected for the decay induced by the high-pass filter) were used to calculate the relative triple-correlation value rel.3C. (given as inset).

## A1.3 Relative triple correlation for ternary complexes of fluorescently tagged proteins

The rel.3C. is a measure of the relative amount of ternary complexes in a system containing three fluorescently tagged protein species. Incomplete maturation or non-fluorescent photophysical states of FP tags will reduce the amount of detectable ternary complexes. To quantify the maximum rel.3C. that can be expected in an experiment, we calculate rel.3C. values for ternary complexes of (i) 1:1:1 or (ii) 2:2:2 stoichiometry, under the assumption that each fluorescent protein can be detected with a probability $P_f$ For simplicity, we assume the same $P_f$ and molecular brightness $\varepsilon$ for all three fluorophore species. Generally, the ACF and 3CF amplitudes for fully formed ternary complexes (i.e., in the absence of partially formed complexes) of concentration $c$ composed of species 1, 2, and 3 with variable stoichiometry $l{:}m{:}n$ are calculated as follows (**Kim et al., 2005**):

$$G_1(0,0) = \frac{c\left(\sum_{i=1}^{l}(i\varepsilon)^2\binom{l}{i}p_f^i(1-p_f)^{l-i}\right)}{V_{eff}\left(c\sum_{i=1}^{l}i\varepsilon\binom{l}{i}p_f^i(1-p_f)^{l-i}\right)^2}$$

(analogously $G_2(0,0)$, $G_3(0,0)$ with upper index $m,n$),

$$G_{3C}(0,0) = \frac{c\left(\sum_{i=1}^{l}(i\varepsilon)^2\binom{l}{i}p_f^i(1-p_f)^{l-i}\right)\left(\sum_{j=1}^{m}(j\varepsilon)^2\binom{m}{j}p_f^j(1-p_f)^{m-j}\right)\left(\sum_{k=1}^{n}(k\varepsilon)^2\binom{n}{k}p_f^k(1-p_f)^{n-k}\right)}{V_{eff}\left(c\sum_{i=1}^{l}i\varepsilon\binom{l}{i}p_f^i(1-p_f)^{l-i}\right)\left(c\sum_{j=1}^{m}j\varepsilon\binom{m}{j}p_f^j(1-p_f)^{m-j}\right)\left(c\sum_{k=1}^{n}k\varepsilon\binom{n}{k}p_f^k(1-p_f)^{n-k}\right)}.$$

From these amplitudes, the rel.3C. can be calculated (see Materials and methods). We obtain rel.3C. = $p_f^2$ = 0.49 (1:1:1 stoichiometry) and rel.3C. = $4p_f^2/(p_f+1)^2 \approx 0.68$ (2:2:2 stoichiometry) for $p_f$ = 0.7. Due to imperfect optical overlap, experimentally detectable rel.3C. values will be lower than these values. To estimate the fraction of ternary complexes than can be detected, we compare experimental rel.cc. values obtained for all FP combinations on a positive control (FP

heterotrimers) in pair-wise cross-correlation analysis with the expected value of rel.cc. = 0.7 for $p_f$ = 0.7 (see Section A1.1). The average rel.cc. value of 0.65 detected for mEGFP and mEYFP signal (see *Figure 6D*) was close to the expected value, hence, almost all complexes containing fluorescent mEGFP and mEYFP were detectable. On the other hand, rel.cc. values for mEGFP and mCherry2 (0.48)/mEYFP and mCherry2 (0.53) were ca. 70% of the expected value (*Figure 6D*). Hence, we estimate that ca. 70% of complexes carrying an mCherry2 tag and an mEGFP or mEYFP tag are detectable due to nonoptimal overlap of excitation/detection volumes. We can therefore assume that for the case of ternary complexes ca. 70% of all fully fluorescent ternary complexes that are present in the sample are optically detectable. The expected experimental rel.3C. values are thus approximately 0.34 and 0.48 for complete binding in 1:1:1 and 2:2:2 stoichiometry, respectively.

