## [Decision Letter]

**Acceptance summary:**

This manuscript provides a new tool to study multi-protein interaction in living cells based on fluorescence fluctuation spectroscopy. The paper includes a reliable validation and description of the method as well as a proof of principle application and assessment of potential limitations.

**Decision letter after peer review:**

Thank you for submitting your article "Multi-color fluorescence fluctuation spectroscopy in living cells via spectral detection" for consideration by *eLife*. Your article has been reviewed by 3 peer reviewers, one of whom is a member of our Board of Reviewers, and the evaluation has been overseen by Anna Akhmanova as the Senior Editor. The following individuals involved in review of your submission have agreed to reveal their identity: Erdinc Sezgin (Reviewer #2); Thorsten Wohland (Reviewer #3).

Essential revisions:

The manuscript is well written and the data very diligently analysed. There remain mainly technical questions and some points that should be discussed to ensure consistency of the results.

1. Figure 1: It seems like the time resolution of the measurements are on the order of several milliseconds. But the diffusion time measured are on the order of ~9 ms. With the time resolution relatively close to the measured diffusion time, did the authors check whether diffusion times are biased?

2. Page 11: Why are correlation amplitudes limited to positive values? The authors mention this also in the supplement but don't explain why that is necessary. For true correlations, only positive values should be obtained. And I assume that is so in their case, if not that should be addressed. Any negative values would be a clear indication that the correlation just indicates noise. This could also be compared with the actual correlation times measured. For correlations of noise, it is not only the amplitude that varies strongly but also the correlation times typically vary widely and mostly do no coincide with expected or reasonable values. The authors might be able to use that criterion to identify non-correlated data and thus avoid the positive amplitudes which are artifacts of the restrictions of the fitting parameters.

3. Page 11 and 12: It would be interesting if the authors could determine whether the observed correlation amplitudes are consistent with the probability of the different FPs (mEGFP, mEYFP, mCherry2 with rel cc amplitudes ranging from 0.45-0.79). If FRET influences these amplitudes, the authors might be even able to extract FRET efficiencies and demonstrate FRET. Another possibility would be to measure fluorescence lifetimes to determine FRET?

4. Page 19/20: The diffusion coefficient for the hetero-tetramer is only about half of the one determined for hetero-dimers. As these are membrane probes with the fluorescent proteins not expected to interact with the membrane, is there an explanation for the factor 2?

5. Did the authors determine the various observation volumes, given the concentrations they measure and the probability of fluorescence of the fluorescent proteins they determined? And is this consistent with the difference in diffusion coefficient they see for the hetero-oligomers?

6. Page 22: The authors indicate that they normalize their RSICS brightness data to measurements of monomers on the same day. It would be interesting if the authors could comment on the day-to-day variability of their calibrations.

7. Figure 3. M2 protein oligomerize but its interacting partner LC3 does not, does that mean each single LC3 associates with multiple M2 protein? Can authors confirm this by looking at the diffusion coefficients from the FCs curves?

8. The authors provided a validation of the method in HEK cells expressing the three Fluorescent Proteins in the plasma membrane in different oligomerization states (Figure 3C). I'm wondering how relatively changes in concentration/expression i.e. fluctuations within the 3 species of probes would influence the observable. Several applications might aim to study interaction within proteins of different abundance, it is important to understand the relative concentration range where the method can be used and provide reliable results. I suggest the authors to provide further experiment and eventually simulation to characterize such dynamic range.

9. The authors applied SFSCS to study how the Influenza A virus matrix protein 2 interact with the autophagy protein LC3 and the tetraspanin CD9. They found that IAV preferentially interact LC3 but not with CD9 (Figure 3). How the position of the label influences the cross-correlation studies?

10. For this technique to be applicable by other researchers, the data analysis tool should be openly available to others. Can authors put their software in an open repository?

11. Since it is a new technique all the limitations of the technique can be discussed in a "limitations" subsection. That would give the readers a clear picture what can and cannot be done with this new technique.

*Reviewer #1 (Recommendations for the authors):*

Multiplexing methods focusing on dynamic studies are challenging yet very important to understand cellular mechanism at the molecular level. Usually such studies are complex and done on specialized system, this work presents a method easily translatable to commercial system and compatible with standard fluorophores with a well-described analysis pipeline.

Therefore, it can open up several new dynamic studies with 3-4 fluorophores simultaneous read-out.

The choice of fluorophores, its dimerization tendency and the relative labeling densities of each species might influence the cross-correlation observable so a careful validation should be considered and discussed to validate the general applicability of the methodology in various biological applications.

To the author:

1 – The authors provided a validation of the method in HEK cells expressing the three Fluorescent Proteins in the plasma membrane in different oligomerization sates (Figure 3C). I'm wondering how relatively changes in concentration/expression i.e. fluctuations within the 3 species of probes would influence the observable. Several applications might aim to study interaction within proteins of different abundance, it is important to understand the relative concentration range where the method can be used and provide reliable results. I suggest the authors to provide further experiment and eventually simulation to characterize such dynamic range.

2 –The authors applied SFSCS to study how the Influenza A virus matrix protein 2 interact with the autophagy protein LC3 and the tetraspanin CD9. They found that IAV preferentially interact LC3 but not with CD9 (Figure 3). How the position of the label influences the cross-correlation studies?

*Reviewer #2 (Recommendations for the authors):*

In this paper, the authors developed a new modality of multi-color FCS using spectral detection to investigate stoichiometry of multi-component complexes in live cells. They showed the proof-of-principle with tandem proteins. Furthermore, they showed the biological application of their methodology by investigating Influenza virus components. This will be a useful addition to live cell spectroscopy tools to study protein-protein interactions. The authors' claims are supported by the data throughput the manuscript. The data is analyzed carefully, and results were reported clearly. This method is likely to be used by cell biologists to determine the stoichiometry of multi-protein complexes.

– Figure 3. M2 protein oligomerize but its interacting partner LC3 does not, does that mean each single LC3 associates with multiple M2 protein? Can authors confirm this by looking at the diffusion coefficients from the FCs curves?

– For this technique to be applicable by other researchers, the data analysis tool should be openly available to others. Can authors put their software in an open repository?

– Since it is a new technique all the limitations of the technique can be discussed in a "limitations" subsection. That would give the readers a clear picture what can and cannot be done with this new technique.

*Reviewer #3 (Recommendations for the authors):*

This manuscript is a carefully conducted study of multi-color fluorescence fluctuation spectroscopy as applied to plasma membranes. By using two-wavelength excitation and spectral detection in a scanning mode the authors show that they can determine the cross-correlation between up to four different probes in a single measurement. In addition, they build on their earlier work and show that the collected data can be analysed by Number and Brightness analysis, providing access to biomolecular stoichiometry. The authors use a range of protein constructs that include between one to four fluorescent proteins, which they use in different compositions to demonstrate that they can analyse all possible interactions of four probes in a single measurement. They then apply the technique to the interaction of influenza A proteins. They show that the influenza A virus (IAV) matrix protein 2 (M2) interacts more strongly with LC3 compared to CD9, both host cell factors. As they measure multiple probes simultaneously, the authors can go beyond binary correlations. Using triple correlations the authors show that they can detect the interaction of the proteins PA, PB1, and PB2 of the IAV polymerase complex. The extension of FCS to four probes, the demonstration of triple correlations and the application to a biological context provides important progress in fluorescence fluctuation spectroscopy and will allow the measurements of complex interactions at the cellular membranes.

The manuscript is well written and the data very diligently analysed. There remain mainly technical questions and some points that should be discussed to ensure consistency of the results.

1. Figure 1: It seems like the time resolution of the measurements are on the order of several milliseconds. But the diffusion time measured are on the order of ~9 ms. With the time resolution relatively close to the measured diffusion time, did the authors check whether diffusion times are biased?

2. Page 11: Why are correlation amplitudes limited to positive values? The authors mention this also in the supplement but don't explain why that is necessary. For true correlations, only positive values should be obtained. And I assume that is so in their case, if not that should be addressed. Any negative values would be a clear indication that the correlation just indicates noise. This could also be compared with the actual correlation times measured. For correlations of noise, it is not only the amplitude that varies strongly but also the correlation times typically vary widely and mostly do no coincide with expected or reasonable values. The authors might be able to use that criterion to identify non-correlated data and thus avoid the positive amplitudes whioch are artefacts of the restrictions of the fitting parameters.

3. Page 11 and 12: It would be interesting if the authors could determine whether the observed correlation amplitudes are consistent with the probability of the different FPs (mEGFP, mEYFP, mCherry2 with rel cc amplitudes ranging from 0.45-0.79). If FRET influences these amplitudes, the authors might be even able to extract FRET efficiencies and demonstrate FRET. Another possibility would be to measure fluorescence lifetimes to determine FRET?

4. Page 19/20: The diffusion coefficient for the hetero-tetramer is only about half of the one determined for hetero-dimers. As these are membrane probes with the fluorescent proteins not expected to interact with the membrane, is there an explanation for the factor 2?

5. Did the authors determine the various observation volumes, given the concentrations they measure and the probability of fluorescence of the fluorescent proteins they determined? And is this consistent with the difference in diffusion coefficient they see for the hetero-oligomers?

6. Page 22: The authors indicate that they normalize their RSICS brightness data to measurements of monomers on the same day. It would be interesting if the authors could comment on the day-to-day variability of their calibrations.

---

## [Author Response]

Essential revisions:The manuscript is well written and the data very diligently analysed. There remain mainly technical questions and some points that should be discussed to ensure consistency of the results.1. Figure 1: It seems like the time resolution of the measurements are on the order of several milliseconds. But the diffusion time measured are on the order of ~9 ms. With the time resolution relatively close to the measured diffusion time, did the authors check whether diffusion times are biased?

The time resolution of the SFSCS measurements was ca. 0.8 ms, since the line-scan time was ca. 0.4 ms and a 2-line binning was performed. To assess whether this time resolution is sufficient to detect the diffusion dynamics observed in the experiments, we re-analyzed a three-species SFSCS measurement acquired on cells co-expressing membrane-anchored FPs (mp-1x-G + mp-1x-Y + mp-1x-Ch2) using a variable time binning (i.e., effective time resolution varying here from ca. 0.4 ms to 6 ms, see Author response image 1). The obtained diffusion times (between 6 and 9 ms at high time resolution, panel A) and particle numbers (panel B) are very similar for the first three binning modalities and start to decrease above 2 ms binning, indicating that a resolution lower than this value might induce biased estimates. Hence, we conclude that the utilized time resolution of less than 1 ms is sufficient to reliably probe the dynamics observed in our samples, and state this now in lines 981-983 of the manuscript. We have furthermore added a short comment on time resolution in the limitations section (lines 827-830).

**Author response image 1. sa2fig1:** Example of a three-species SFSCS analysis for a measurements on cells co-expressing mp-mEGFP, mp-mEYFP, and mp-mCherry2 with variable time binning of [1,2,4,8] x T, acquired with a line-scan time of T~0.4 ms.

2. Page 11: Why are correlation amplitudes limited to positive values? The authors mention this also in the supplement but don't explain why that is necessary. For true correlations, only positive values should be obtained. And I assume that is so in their case, if not that should be addressed. Any negative values would be a clear indication that the correlation just indicates noise. This could also be compared with the actual correlation times measured. For correlations of noise, it is not only the amplitude that varies strongly but also the correlation times typically vary widely and mostly do no coincide with expected or reasonable values. The authors might be able to use that criterion to identify non-correlated data and thus avoid the positive amplitudes which are artifacts of the restrictions of the fitting parameters.

We apologize for the misunderstanding: The limitation to positive values does not stem from specific boundary settings in the fitting procedure (as it might be indeed inferred by what was written in lines 854 and 248). We have now clarified these passages (lines 278-279, 1024-1029). Briefly, the presence of positive correlation values in noisy curves is a consequence of positive starting parameters for the fit routine and, probably, its converging to a local minimum (i.e., to a positive correlation value).

The (practical) reason for consistently using positive starting values for the correlation was that in the case of clear interactions (e.g. cross-correlation for the positive controls), only positive initial amplitude values resulted in robust and reproducible fitting of the CCF. Negative initial values resulted in errors in the fit routine or fit curves that clearly did not provide a reliable fit to the CFs. To perform an unbiased analysis, the same (positive) initial values were thus used for all samples, sometimes resulting in low “false-positive” amplitudes in the absence of interactions for noisy curves. This in now stated in lines 1024-1029.

We agree with the reviewer that, for the case of pure noise, a symmetric distribution of fit amplitudes around 0 is expected. We have thus implemented an alternative fit routine, which calculates the starting parameter for the amplitude as the average of the first 5 points of each CF. In the case of pure noise, this results in positive or negative fit amplitudes, as mentioned by the reviewer. In the case of interactions, the fit converges to positive values. To evaluate this fit routine, we have reanalyzed a set of three-species SFSCS measurements on cells co-expressing mp-mCherry2-mEGFP hetero-dimers and mp-mEYFP. The new fit procedure (“fit routine 2” in Author response image 2) indeed results in amplitude values that scatter around 0 for non-interacting species (e.g. G,Y; Y,Ch2). This confirms that the previously obtained low cross-correlation does not result from artefacts in the spectral filtering. On the contrary, positive values are obtained for truly correlating samples (e.g. G,Ch2) and they are similar to the results shown in the first version of the manuscript.

We furthermore agree with the reviewer that unreasonable values of the cross-correlation diffusion time can be used to identify correlation of noise. When we apply a threshold of 5 times the maximum diffusion time obtained from ACFs (i.e., considering higher values as an artefact of noise and setting the CCF fit amplitudes of these measurements to zero, for the respective FP combination), some likely false-positive correlation is removed (“fit routine 1, filtered” in Author response image 2). Nevertheless, this procedure requires to set an arbitrary threshold for filtering of the diffusion times, which may be subjective and depend on the specific sample dynamics. Overall, the differences obtained for the different fit procedures are small. For this reason and to keep the analysis comparable with our previously published work, we prefer not to modify the already proposed analysis. Nevertheless, we have added a supplementary figure (Figure 3—figure supplement 2) demonstrating the different fit options so that readers can choose the type of analysis best suited for their experiment. We refer to this figure in the results (lines 278-279). We thank the reviewer for bringing up this important issue.

**Author response image 2. sa2fig2:** Relative cross-correlation for three-species SFSCS experiments described in Figure 3 of the manuscript, analyzed using different fit routines. CCFs obtained from measurements on cells co-expressing mp-Cherry2-mEGFP hetero-dimers and mp-mEYFP were either fitted with the same positive initial value for the amplitude (fit routine 1), or with the average of the first five points of each CCF (fit routine 2). For non-correlated data (e.g. G,Y and Y,Ch2 combinations), the second fit routine may converge to negative fit amplitudes, resulting in a distribution of rel.cc. values scattered around 0. Fit routine 1 always converged to positive amplitude values, causing low residual false-positive rel.cc. values. Filtering based on “unreasonable” cross-correlation diffusion time may remove some of the residual positive rel.cc. (fit routine 1, filtered). Here a threshold value of five times the maximum of the two diffusion times obtained from ACFs for each respective FP combination was chosen. For correlated data, e.g. G,Ch2, both fit routines converged to comparable positive values.

3. Page 11 and 12: It would be interesting if the authors could determine whether the observed correlation amplitudes are consistent with the probability of the different FPs (mEGFP, mEYFP, mCherry2 with rel cc amplitudes ranging from 0.45-0.79). If FRET influences these amplitudes, the authors might be even able to extract FRET efficiencies and demonstrate FRET. Another possibility would be to measure fluorescence lifetimes to determine FRET?

For FPs with a probability p_f_ to be fluorescent, the expected rel.cc. is equal to p_f_, as we discuss in the Appendix of the paper (Appendix-table 1). Since all FPs (mEGFP, mEYFP, mCherry2) exhibit similar p_f_ of ≈70%, the expected rel.cc. is ≈0.7 for all FP pairs. The values obtained for cross-correlation of mEGFP and mEYFP signals (with a rigid linker) are indeed close to the expected value (e.g. rel.cc.=0.72 for mEYFP-mEGFP hetero-dimers in 2-species SFSCS, Figure 1). A significantly lower rel.cc. of 0.60 was obtained with a short linker peptide between the two FPs, suggesting the presence of FRET (see Figure 1—figure supplement 3). For the other FP combinations requiring excitation with two laser lines, the obtained rel.cc. values are lower than expected (e.g. rel.cc.=0.56 and 0.57 for mEGFP/mCherry2 and mEYFP/mCherry2 in heterotrimers). In addition to non-fluorescent states, differences in observation volumes and FRET can reduce the experimentally observed rel.cc., as pointed out by the reviewer. We address the issue of imperfect overlap below, in the response to question 5. To minimize FRET artifacts, FPs in hetero-trimers and tetramers were linked by a rigid linker. Nevertheless, residual FRET may still occur. To assess FRET between FPs, we quantified the molecular brightness of FPs coupled in hetero-oligomers and compared it with the brightness obtained on the same FPs expressed as independent monomers. The brightness ratios q_i_ (i=G,Y,Ch2) are shown below for SFSCS measurements on cells expressing mp-mEYFP-mCherry2-mEGFP hetero-trimers (Author response image 3) or coexpressing mp-mCherry2-mEGFP hetero-dimers and mp-mEYFP (Author response image 4). FRET efficiencies (f_E_) are related to the decrease in donor brightness, f_E_=1-q_G_ (Foo et al., BJ, 2012). We observed a systematic reduction of the brightness of mEGFP, no (Author response image 4) or small (Author response image 3, hetero-trimer) changes of the mEYFP brightness, and a consistent increase of mCherry2 brightness. This indicates residual FRET between, e.g. mEGFP donor and mCherry2 acceptor. Based on the experimental q values, we can calculate the expected rel.cc. assuming p_f_≈0.7 for all 3 FPs, using the formulas derived in Foo et al., BJ, 2012. We obtain rel.cc. values of 0.63 and 0.69 for the FP combination most affected by FRET (mEGFP/mCherry2), for hetero-dimers and hetero-trimers respectively. Thus, FRET has a minor effect (<10%) on the cross-correlation and alone cannot explain the reduced rel.cc. observed for pairs of green/yellow and red FPs. It should be noted that, in the case of hetero-trimers, our analysis based on the effective molecular brightness is a simplified approximation, since all three FP pairs may undergo FRET to different extent. We agree with the reviewer that it is important to keep FRET into consideration when planning experiments and constructs, in particular because hetero-oligomers are commonly used as a positive control to normalize the rel.cc. obtained on proteins of interest. We have therefore added a comment on FRET to the Limitations section (lines 864-869).

**Author response image 3. sa2fig3:** Molecular brightness obtained from three-species SFSCS measurements on mp-mEYFP-mCherry2-mEGFP hetero-trimers, normalized to the values obtained on cells co-expressing mp-mEGFP, mp-mEYFP and mp-Cherry2.

**Author response image 4. sa2fig4:** Molecular brightness obtained from three-species SFSCS measurements on cells co-expressing mpmCherry2-mEGFP heterodimers and mp-mEYFP, normalized to the values obtained on cells co-expressing mpmEGFP, mp-mEYFP and mp-Cherry2.

4. Page 19/20: The diffusion coefficient for the hetero-tetramer is only about half of the one determined for hetero-dimers. As these are membrane probes with the fluorescent proteins not expected to interact with the membrane, is there an explanation for the factor 2?

We think that this comment might arise from a misunderstanding. The data shown on Page 19/20 were obtained by RSICS measurements in the cytoplasm, not on the plasma membrane. Thus, diffusion coefficients reasonably depend on oligomer size. The obtained values for the diffusion of FP oligomers, e.g. diffusion of FP tetramer about 2.5 times slower than diffusion of monomers, are in good agreement with previous studies (e.g. Dross, PlosOne, 2009, Erdel Nat. Comm. 2014, Dunsing, Sci. Rep., 2018). We have clarified this issue in lines 447 and 467.

5. Did the authors determine the various observation volumes, given the concentrations they measure and the probability of fluorescence of the fluorescent proteins they determined? And is this consistent with the difference in diffusion coefficient they see for the hetero-oligomers?

We have already taken imperfect overlap of the observation volumes into account for comparison of the experimental triple correlation with simulation results (see results and Appendix for triple correlation) but have not determined the size and potential displacement of observation volumes explicitly.

As mentioned in our response to point 3, differences in observation volumes for different FP channels may cause the lower than expected rel.cc. obtained for pairs of green/yellow and red FPs, e.g. mEGFP/mCherry2. To evaluate such differences, we have analyzed the relative ACF amplitudes and diffusion times obtained from SFSCS measurements on FP hetero-oligomers, i.e. mp-mEYFP-mCherry2-mEGFP hetero-trimers and mp-mEGFP-mCherry2 hetero-dimers in the different channels (e.g. G,Y,Ch2 and G,Ch2). As shown in Author response image 5, the data spread for the diffusion times of the hetero-trimer is too large to draw definitive conclusions. Therefore, we have focused on mp-mCherry2-mEGFP hetero-dimers and started evaluating the relative ACF amplitudes. In addition to differences in observation volumes, the amplitudes are also affected by p_f_ (e.g. lower p_f_ corresponds to higher amplitude). For all FPs evaluated here, p_f_ values are similar (ca. 0.7, see RSICS data in Figure 6 and previously published data in Dunsing et al., Sci. Rep., 2018). Thus, if the observation volumes were the same, the relative amplitudes should be close to 1, as shown indeed in Author response image 6. This indicates that differences in observation volumes or p_f_ are small. More precisely, one would expect a slightly larger observation volume for mCherry2 than for mEGFP (due to the higher wavelength) and thus an amplitude ratio G_Ch2_/G_G_ smaller than 1. However, values slightly larger than 1 are obtained. This might be due to e.g. slightly smaller p_f_ of mCherry2. In addition, residual FRET (see point 3) may increase the ACF amplitude of the acceptor fluorophore, here mCherry2 (see Foo et al., BJ, 2012).

Due to the difficulties in interpreting the ACF amplitudes, we have therefore directly evaluated the diffusion times of mp-mCherry2-mEGFP hetero-dimers obtained from ACFs and from CCFs. In scanning FCS, the diffusion times scale with the effective observation area A_eff_=Π∙S∙w_0_^2^. The results are shown in Author response image 7. The significant increase in diffusion time for mCherry2 (taud_Ch2_/taud_G_=1.35) indicates a ca. 35% difference in size and a slight shift of observation volumes (since the diffusion time obtained from the CCF does not fall in between the values obtained from ACFs for taud_Ch2_ and taud_G_, see the discussion in Foo et al., 2012). Calculating the effective overlapping observation area from the cross-correlation diffusion time (taud_G,Ch2_=1.38*taud_G_), the determined FRET efficiencies (see q factors in response to point 3, q_G_=0.8, q_R_=1.3) and keeping the p_f_ values (0.7 for both) into account, we calculated an expected rel.cc. of ca. 0.50 for mp-mCherry2-mEGFP hetero-dimers, which is very close the experimental value of 0.45. Thus we conclude that the deviation from the expected rel.cc. (solely based on p_f_, i.e. ca. 0.7) for FP combinations involving excitation by two laser lines in general might be caused by both, FRET, and imperfect overlap of the observation volumes.

We have added the issue of imperfect overlap, which applies to SFSCS as to standard FCCS as soon as a second excitation line is used, to the limitations section (lines 824-826) and state it more clearly also in the discussion (lines 670-674). Therein, we stress the point that for two-species measurements, the fact that only one laser line is needed provides a clear advantage of the spectral approach, resulting in higher rel.cc. (up to 0.8).

**Author response image 5. sa2fig5:** Diffusion times obtained from ACFs in three-species SFSCS measurements on cells expressing mpmEYFP-mCherry2-mEGFP hetero-trimers.

**Author response image 6. sa2fig6:** Relative ACF amplitudes obtained from three-species SFSCS measurements on cells co-expressing mp-mCherry2-mEGFP hetero-dimers and mp-mEYFP. Only the amplitude ratio for mEGFP and mCherry2 ACFs is shown.

**Author response image 7. sa2fig7:** Diffusion times obtained from ACFs and CCFs in three-species SFSCS measurements on cells coexpressing mp-mCherry2-mEGFP hetero-dimers and mp-mEGFP. The diffusion times obtained from mEGFP/mCherry2 ACFs and from the CCF are shown here.

6. Page 22: The authors indicate that they normalize their RSICS brightness data to measurements of monomers on the same day. It would be interesting if the authors could comment on the day-to-day variability of their calibrations.

We thank the reviewer for this comment and agree that data on day-to-day variability are highly useful and often underreported in brightness studies. We have examined both the day-to-day variability of the relative brightness values of the homo-dimer control and of the absolute brightness of the monomeric reference on the same day. The data has been added to the SI and refer to in the results (lines 591-592).

7. Figure 3. M2 protein oligomerize but its interacting partner LC3 does not, does that mean each single LC3 associates with multiple M2 protein? Can authors confirm this by looking at the diffusion coefficients from the FCs curves?

Yes, we think that LC3 monomers bind to M2 oligomers in (apparent) average stoichiometry of 1:2 to 1:4. In the discussion of the paper, we speculate already that LC3 does not show higher oligomerization (or higher stoichiometry in the binding to M2) because endogenous LC3 or other cellular proteins might block additional binding sites in M2 oligomers, or because of steric hindrance. We have evaluated the diffusion dynamics of M2 and LC3 (see Author response image 8). Although the rel.cc. between M2 and LC3 is similar to the positive control (indicating that most LC3 is present in complexes with M2), the diffusion dynamics of LC3 are consistently faster (0.03± 0.01 ms) than the dynamics of M2 (0.06±0.01 ms). This is to be expected, since LC3 is a membrane associated protein and its apparent diffusion dynamics is likely influenced by the dynamics of association/dissociation rates (in addition to actual diffusion at the PM). Therefore, we think that the apparent diffusion dynamics of LC3 are difficult to interpret and cannot be used directly to explore M2-LC3 interactions. We have also analyzed the diffusion times obtained from the LC3-M2 CCFs. They show a fairly large variation due to noise but the majority of data points report similar diffusion dynamics (0.07±0.06 ms) as M2 oligomers, which would be expected for binding of a membrane associated to a transmembrane protein.

**Author response image 8. sa2fig8:** Diffusion dynamics determined in three-species SFSCS measurements on HEK 293T cells coexpressing CD9-mEGFP, LC3-mEYFP, and M2-mCh2 (related to Figure 3D-G).

8. The authors provided a validation of the method in HEK cells expressing the three Fluorescent Proteins in the plasma membrane in different oligomerization sates (Figure 3C). I'm wondering how relatively changes in concentration/expression i.e. fluctuations within the 3 species of probes would influence the observable. Several applications might aim to study interaction within proteins of different abundance, it is important to understand the relatively concentration range where the method can be used and provide reliable results. I suggest the authors to provide further experiment and eventually simulation to characterize such dynamic range.

We thank the reviewer for this comment and agree that the relative concentration range that provides reliable results is important to explore. We have therefore already carefully evaluated the SNR of two-species SFSCS measurements as a function of the relative concentrations (i.e. intensities), which we show in Figure 1. In addition, we have extensively discussed the allowed relative concentration range for 2-,3-, and 4-species measurements in the discussion and methods section of the paper (e.g. 1:5 for neighboring FP species in 4-species measurements), and have compared it to the limitations of similar approaches such as fluorescence lifetime correlation spectroscopy. In order to explicitly address the question for three-species SFSCS (e.g. Figure 3), we have now analyzed the SNR of the ACFs obtained from measurements on cells co-expressing mp-mEGFP, mp-mEYFP and mp-mCherry2 (see Author response image 9). The SNR is plotted as a color code in 2-d graphs, for each species as a function of the relative intensity to the other two species. Similar to two-species SFSCS, the SNR of mEYFP (panel B) is mostly compromised, by both signal from mEGFP (horizontal axis) and mCherry2 (vertical axis), i.e. a 6-fold drop of SNR for relative signals ranging from ca. 6:1 (yellow points) to ca. 1:6 (dark blue points). On the contrary, mCherry2 (3-fold change of SNR, panel C) and mEGFP (2-fold change of SNR, panel A) are only moderately affected. As expected, the SNR ratio of mCherry2 ACFs (panel C) depends stronger on mEYFP (vertical axis) than on mEGFP signal (horizontal axis), which is caused by the higher overlap with mEYFP emission. We have added the figure as supplement (Figure 3—figure supplement 3) and refer to it now in the discussion (lines 696-697, 728, 730, 734).

Given that mEYFP is the most compromised of the three species, we have then evaluated the diffusion time and molecular brightness obtained for mEYFP in three groups corresponding to the highest, middle and lowest third of the 31 data points, sorted by SNR. The obtained results (Author response image 1) are very similar and do not show significant differences in the average values or variation of parameter estimates. This shows that the SFSCS analysis works reliably in the here explored concentration range of 1:6 to 6:1 for neighboring FP channels. As we state in the discussion, the relative concentration range may depend on the specific sample, fluorophores and detection efficiency, and thus needs to be determined carefully. We have added an additional comment on this issue to the new limitations section (lines 833-841).

**Author response image 9. sa2fig9:** Noise analysis of three-species SFSCS measurements described in Figure 3. Plotted is the SNR (color coded) of ACFs for mEGFP (A), mEYFP (B), and mCherry2 (C) obtained from SFSCS measurements on HEK 293T cells co-expressing mp-mEGFP, mp-mEYFP, and mCherry2, as a function of the relative signal to the other two FP species. Date were pooled form two independent experiments in which 31 cells were measured in total.

**Author response image 10. sa2fig10:** Diffusion time (left) and normalized brightness obtained for mEYFP from three-species SFSCS measurements on cells co-expressing mp-mEGFP, mp-mEYFP and mp-mCherry2, pooled in tree groups of data points comprised of the lower, middle and upper third of data points, sorted by SNR. The brightness was normalized to the average value of all 31 measurements.

9. The authors applied SFSCS to study how the Influenza A virus matrix protein 2 interact with the autophagy protein LC3 and the tetraspanin CD9. They found that IAV preferentially interact LC3 but not with CD9 (Figure 3). How the position of the label influences the cross-correlation studies?

To minimize the influence of the fluorophore tags on our interaction study, we chose a reasonable labeling approach. The M2 protein was tagged at the extracellular N-terminus. Binding with LC3 occurs via an LIR-motif in the flexible cytoplasmic tail of M2 (Beale et al., Cell Host and Microbe 2014). LC3 docks with the LIR motif via hydrophobic pockets located in the middle region of its sequence (Hamacher-Brady and Brady, Cell. and Mol. Life Sciences, 2016). We placed the mEYFP tag at the C-terminus of LC3, i.e. at some distance from the binding region. Thus, we do not expect interference of the FP tags with M2-LC3 interactions.

For CD9, both N- and C-terminus and intracellular. The mEGFP-tag is located at the C-terminus, with a 10aa linker. Having the tags on M2 and CD9 on different sides of the plasma membrane (EC/IC) should minimize potential hindrance in the interactions. Binding e.g. via the TMDs should not be affected. Given the flexibility of the cytoplasmic tail of M2, we do not think that the tag on CD9 hinders M2-CD9 interactions. Nevertheless, we cannot rule this out completely and thus added a sentence to the manuscript (lines 776-778). In addition, we have added a discussion on the general requirements of fluorophore tags and potential interference with protein-protein interactions to the Limitations section (lines 848-853).

10. For this technique to be applicable by other researchers, the data analysis tool should be openly available to others. Can authors put their software in an open repository?

We agree that it would be very beneficial to have the code openly available. We have therefore uploaded all analysis code (SFSCS, RSICS, and TRICS) and a documentation to run the main scripts on GitHub (https://github.com/VaDu8989/SpectralFFS).

11. Since it is a new technique all the limitations of the technique can be discussed in a "limitations" subsection. That would give the readers a clear picture what can and cannot be done with this new technique.

We thank the reviewer for this idea and have added a “Limitations” subsection at the end of the discussion, summarizing the main technical and conceptual limitations of the technique.